# Recruitment of Mre11 to recombination sites during meiosis

Priyanka Priyadarshini [1] ✉, Mahesh Survi[1], Wael El Yazidi Mouloud[2,3], Regina Bohn[4,5], Steven Ballet [2], Neil Hunter [4,5], Alexander N. Volkov[3,6] & Corentin Claeys Bouuaert [1] ✉

The Mre11 nuclease is part of the conserved MRX complex involved in DNA double-strand break (DSB) repair. During meiosis in budding yeast, MRX is also required for Spo11-mediated programmed DSB formation to initiate homologous recombination. Recruitment of Mre11 to meiotic DSB sites depends on Rec114-Mei4 and Mer2, proposed to organize the DSB machinery via biomolecular condensation. Here, we show that Mre11 and MRX complexes form DNA-dependent, hexanediol-sensitive condensates in vitro. In vivo, Mre11 assembles into DNA damage-dependent foci during mitosis and DSB-independent foci during meiosis. Both in vitro condensates and in vivo foci require Mre11 C-terminal intrinsically-disordered region (IDR). While dispensable for vegetative DNA repair, Mre11 IDR is essential during meiosis, where it mediates interaction with Mer2 via a short α-helix and contains a SUMO-interacting motif that enhances Mre11 recruitment and DSB formation. Together, these findings provide insights into the biophysical properties of Mre11 and its role in initiating meiotic recombination.

Cells continuously experience exogenous and endogenous stress, leading to cytotoxic lesions such as DNA double-strand breaks (DSBs)[1,2]. If not repaired promptly and accurately, DSBs can compromise genome stability, resulting in cellular dysfunction, tumorigenesis, or cell death[3]. While DSBs are generally considered detrimental for genome integrity, several instances of programmed DSBs that serve a physiological purpose are also well-documented, including V(D)J recombination during lymphocyte development, immunoglobulin class-switching, and meiotic recombination[4].

During meiosis, recombination initiated by programmed DNA DSBs serves two critical functions: it ensures accurate segregation of genetic material by forming physical connections between homologous chromosomes and increases genetic diversity through allelic shuffling[5,6]. Meiotic DSBs are generated by the highly conserved type II topoisomerase-like protein Spo11 that acts in a consortium with several accessory factors, referred to as DSB proteins[7,8].

In Saccharomyces cerevisiae, the nine known DSB proteins include Rec114-Mei4 and Mer2 (RMM). These are proposed to assemble into nucleoprotein condensates along meiotic chromosomes[9] that act as sub-cellular compartments that recruit other DSB proteins to stimulate DNA cleavage by controlling Spo11 dimerization[10–14]. This condensation model provides a mechanism that ties DSB formation to the loop-axis organization of meiotic chromosomes[15,16].

DSB formation in S. cerevisiae also depends on the Mre11-Rad50-Xrs2 (MRX) complex[17–20]. In addition, MRX initiates the resection of DSB ends in both meiotic and mitotically cycling cells, through the endonuclease and exonuclease activities of Mre11[21–27]. Further, the MRX complex has notable roles in telomere maintenance, stabilization of replication forks, and viral infection[28–30].

[1]Louvain Institute of Biomolecular Science and Technology, Université catholique de Louvain, Louvain-La-Neuve, Belgium. [2]Research Group of Organic Chemistry, Vrije Universiteit Brussel (VUB), Pleinlaan 2, Brussels, Belgium. [3]Jean Jeener NMR Centre, Vrije Universiteit Brussel (VUB), Pleinlaan 2, Brussels, Belgium. [4]Howard Hughes Medical Institute, University of California Davis, Davis, CA, USA. [5]Department of Microbiology & Molecular Genetics, University of California Davis, Davis, CA, USA. [6]VIB-VUB Center for Structural Biology, VIB, Pleinlaan 2, Brussels, Belgium. ✉e-mail: priyanka.priyadarshini107@gmail.com; corentin.claeys@uclouvain.be

While the roles of MRX in genome stability have been studied extensively, its role in meiotic DSB formation remains poorly understood. Here, we show that Mre11 forms dynamic condensates in vitro, dependent on its C-terminal disordered tail. The Mre11 C-terminus was previously shown to be required for meiotic DSB formation[31,32] and was recently found to bind Spo11[33]. We demonstrate that the C-terminus of Mre11 also binds directly to Mer2 and contains a SUMO-interacting motif that also facilitates its recruitment to DSB sites. Thus, our work delineates multiple mechanisms whereby the Mre11 C-terminal tail specifically promotes meiotic DSB formation.

## Results

### Mre11 and the MRX complex assemble DNA-dependent condensates in vitro

During meiosis and in mitotically cycling cells exposed to DNA damage, budding yeast Mre11 forms numerous foci visible by immunofluorescence microscopy[32,34,35]. In vitro, Mre11-Rad50 complexes have been shown to form higher-order assemblies on DNA, mediated by Rad50 oligomerization[36]. However, whether Mre11 itself participates in the assembly of higher-order structures has not been established.

To investigate the biochemical properties of Mre11 and the MRX complex, we purified Alexa488-labeled Mre11 and MRX complexes with an eGFP tag at the N-terminus of Mre11 (Supplementary Figs. 1a, 2a) and asked whether they undergo condensation in vitro. In the presence of plasmid DNA, we found that Mre11 and MRX form large clusters visible by fluorescence microscopy (Fig. 1a, c). While foci were almost absent when DNA was omitted, low concentrations of DNA led to few foci of very high intensity, suggesting that the limited number of DNA molecules provide nucleation points for Mre11 to agglomerate (Supplementary Fig. 1b). Time-course analysis showed a progressive increase in foci number and intensity over time (Supplementary Fig. 1c), and protein titration revealed a steep increase in foci intensity above 200 nM Mre11 (Supplementary Fig. 1d). A similar concentration-dependent increase in foci intensity was also observed for MRX condensates (Supplementary Fig. 2b). Mre11 and MRX condensation was also promoted in the presence of crowding agents (Fig. 1e, f and Supplementary Fig. 2b). Mre11 condensation was enhanced in the presence of divalent metal ions, likely through charge neutralization (Supplementary Fig. 1e).

To address whether the condensates are reversible, we assembled the condensates for 20 min and treated the reactions with DNase I. While nuclease treatment strongly reduced the numbers of Mre11 and MRX foci (Fig. 1b, d), the minority of Mre11 foci that resisted nuclease treatment tended to accumulate more protein (Supplementary Fig. 1f), although such accumulation of proteins in nuclease-resistant condensates was not observed with MRX (Fig. 1d). These observations are consistent with the idea that, following initial nucleation by DNA, condensates grow primarily through protein-protein interactions.

To further test this interpretation, we asked whether Mre11 and MRX condensates are sensitive to 1,6-hexanediol, an aliphatic alcohol that often dissolves biomolecular condensates[37]. Ten-minute treatment with 5% hexanediol led to a strong reduction in foci intensity (Fig. 1g, I). Similarly, brief exposure to 500 mM NaCl strongly reduced foci intensity (Fig. 1h, j). Hence, Mre11 and MRX nucleoprotein condensates are stabilized through a combination of ionic and weak hydrophobic interactions.

### The C-terminal IDR of Mre11 is required for condensation

Mre11 consists of an N-terminal phosphodiesterase domain, a capping domain followed by a Rad50 binding domain, and a C-terminal tail, predicted to be largely disordered (Fig. 2a, b)[38–40]. The last 15 residues form a short, conserved helix. It was previously shown that the C-terminal 49 amino acids are dispensable for mitotic DSB repair but important for Mre11 binding to meiotic DSB hotspots[41] and for DSB formation[31].

To identify the domain(s) required for Mre11 condensation, we purified eGFP-tagged truncations (Supplementary Fig. 3a, b). Deletion of residues 10–270 that removes most of the phosphodiesterase domain and residues 290–472 that removes the capping domain and most of the Rad50-binding domain led to reduced numbers of foci that retained high intensity (Fig. 2c). In contrast, deleting the disordered region (residues 524–677, $\Delta IDR$) or the C-terminal 49 residues (residues 644–692, $\Delta C49$) strongly reduced Mre11 foci intensity (Fig. 2d). Similarly, the Mre11 C-terminal extremity was also required for efficient focus formation of eGFP-tagged MRX complexes (Fig. 2e).

To understand the effects of Mre11 truncations on DNA-binding activity, we quantified binding to a pUC19 plasmid substrate by gel shift analysis (Supplementary Fig. 3c, d). All of the truncations retained significant DNA-binding activity. Similar to Mre11$^{WT}$, Mre11$^{\Delta10-270}$ and Mre11$^{\Delta290-472}$ produced fast-migrating complexes at concentrations up to ~25 nM protein, then formed complexes of reduced mobility at higher protein concentration, presumably indicative of higher-order assemblies (supershifts). However, these species of reduced electrophoretic mobility were absent in Mre11$^{\Delta IDR}$ and Mre11$^{\Delta C49}$.

While the Mre11 IDR is required for efficient Mre11 condensation, purified eGFP-tagged Mre11$^{IDR}$ alone failed to form condensates and showed reduced DNA binding as compared to the wild type or Mre11$^{\Delta IDR}$ (Supplementary Fig. 3e, f). We conclude that, while protein-DNA interactions are important for the formation of Mre11 foci in vitro, multivalent protein-protein interaction through low-complexity regions contributes to condensation.

### In mitotically cycling cells, the Mre11 IDR is required for focus formation but dispensable for DNA repair

To investigate the relationship between DNA-damage-induced Mre11 foci and DNA repair, we treated vegetative yeast cells expressing myc-tagged Mre11 with methylmethanesulfonate (MMS) and visualized focus formation by immunofluorescence microscopy. A brief 4-5 min exposure to 5% 1,6-hexanediol readily dissolved MMS-induced foci, which instead coalesced into a few aggregates of high intensity (Fig. 3a, b).

Deletion of the IDR (mre11-ΔIDR) or C-terminal 49 residues (mre11-ΔC49) severely reduced MMS-induced focus formation relative to wild type (Fig. 3c). Despite being important for focus formation, the C-terminus of Mre11 was previously shown to be dispensable for vegetative DNA repair[31]. Consistently, the mre11-ΔC49 mutant was resistant to treatment with MMS or camptothecin (CPT). Similarly, mre11-ΔIDR mutants were as resistant to MMS or CPT as wild-type cells (Fig. 3d). Hence, the formation of condensate-like Mre11 foci is dispensable for DSB repair in mitotically cycling cells.

### Mer2 condensates directly recruit Mre11 during meiosis

During meiosis, Mre11 appears as discrete foci in immunofluorescence staining of prophase I chromosome spreads[32,33]. Analogous to mitotic cells, a brief exposure to 1,6-hexanediol disassembles these foci (Fig. 4a and Supplementary Fig. 4a). However, unlike vegetative conditions, we did not detect large Mre11 aggregates upon 1,6-hexanediol treatment during meiosis.

While Mre11 foci in mitotic cells are formed in response to DNA damage, meiotic Mre11 foci are formed independently of DSBs in cells carrying the catalytically inactive spo11-Y135F allele[42] (Supplementary Fig. 4d, e). In a wild-type background, the number of Mre11 foci tended to decrease in late meiotic prophase, while foci accumulated in a spo11-Y135F mutant (Supplementary Fig. 4d). Consistent with the known meiotic function of the Mre11 C-terminus[31], mre11-C49 cells produced much fewer foci in meiosis than the wild-type MRE11 cells (Fig. 4b). However, the few foci observed had an intensity similar to the wild type, suggesting that in vivo, the Mre11 C-terminus is important for the nucleation of Mre11 foci, but not their growth. In contrast, the mre11-ΔIDR strain produced fewer and much weaker foci. Approximately 40%

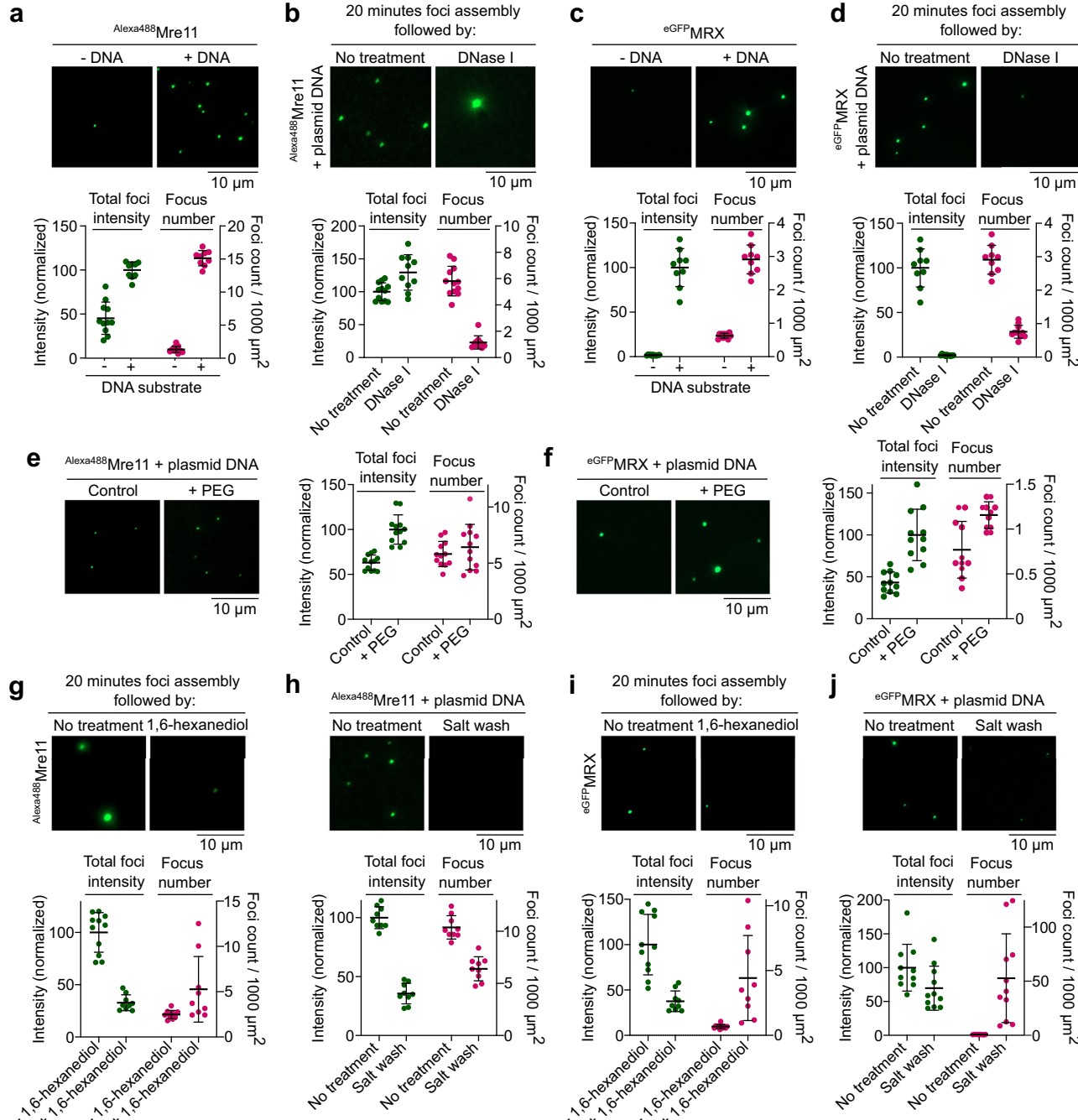

**Fig. 1 | Mre11 and the MRX complex assemble DNA-dependent condensates in vitro. a**, **c** Effect of the presence of plasmid DNA on condensation of (**a**) Alexa488-labeled Mre11 (*n* = 11 and 9 fields of view for reactions without and with DNA, respectively) and (**c**) eGFP-tagged MRX (*n* = 11 (- DNA) and 9 (+ DNA)). Experiments were performed without PEG. **b**, **d** Effect of DNase I, in the absence of PEG, on (**b**) Mre11 (*n* = 9 (no treatment) and 11 (DNase I)) and (**d**) MRX (*n* = 9 (no treatment) and 11 (DNase I)) condensates. Condensates were pre-assembled for 20 min with 5.7 nM pUC19 prior to challenge. Quantification represents total foci intensity (green) in a field of view and total number of foci per 1000 μm² (magenta). Foci intensity is normalized to the mean intensity of samples with DNA for (**a**, **c**) and untreated samples for (**b**, **d**). Error bars represent mean ± SD from the indicated (*n*) fields of view. **e**, **f** Effect of the presence of 5% PEG-3350 on (**e**) Mre11 (*n* = 12) and (**f**) MRX (*n* = 11) condensate formation. Quantification represents total foci intensity (green) in a field of view (normalized to the mean of + PEG images) and total

number of foci per 1000 μm² (magenta). Error bars represent mean ± SD from the indicated (*n*) fields of view. **g**–**j** Effect of challenge with 5% 1,6-hexanediol and 0.5 M NaCl on (**g**, **h**) Mre11 and (**i**, **j**) MRX condensates. Condensates were assembled for 20 min prior to challenge and incubated for 15 min and 10 min, respectively, after challenge. In the control reaction, an equivalent volume (1/20) of water was added instead of hexanediol. Quantification shows foci intensity (normalized to the mean of untreated samples) and total number of foci per 1000 μm² (magenta). In (**g**, **i**), untreated samples have *n* = 11 (Mre11) and *n* = 11 (MRX), and hexanediol-treated samples have *n* = 10 (Mre11) and *n* = 9 (MRX). In (**h**, **j**), untreated samples have *n* = 9 (Mre11) and *n* = 10 (MRX), and NaCl-treated samples have *n* = 10 (Mre11) and *n* = 11 (MRX). Error bars represent mean ± SD from the indicated (*n*) fields of view. The foci count for untreated MRX foci is between 0.83 – 1.33 foci per 1000 μm². Source data are provided as a Source data file.

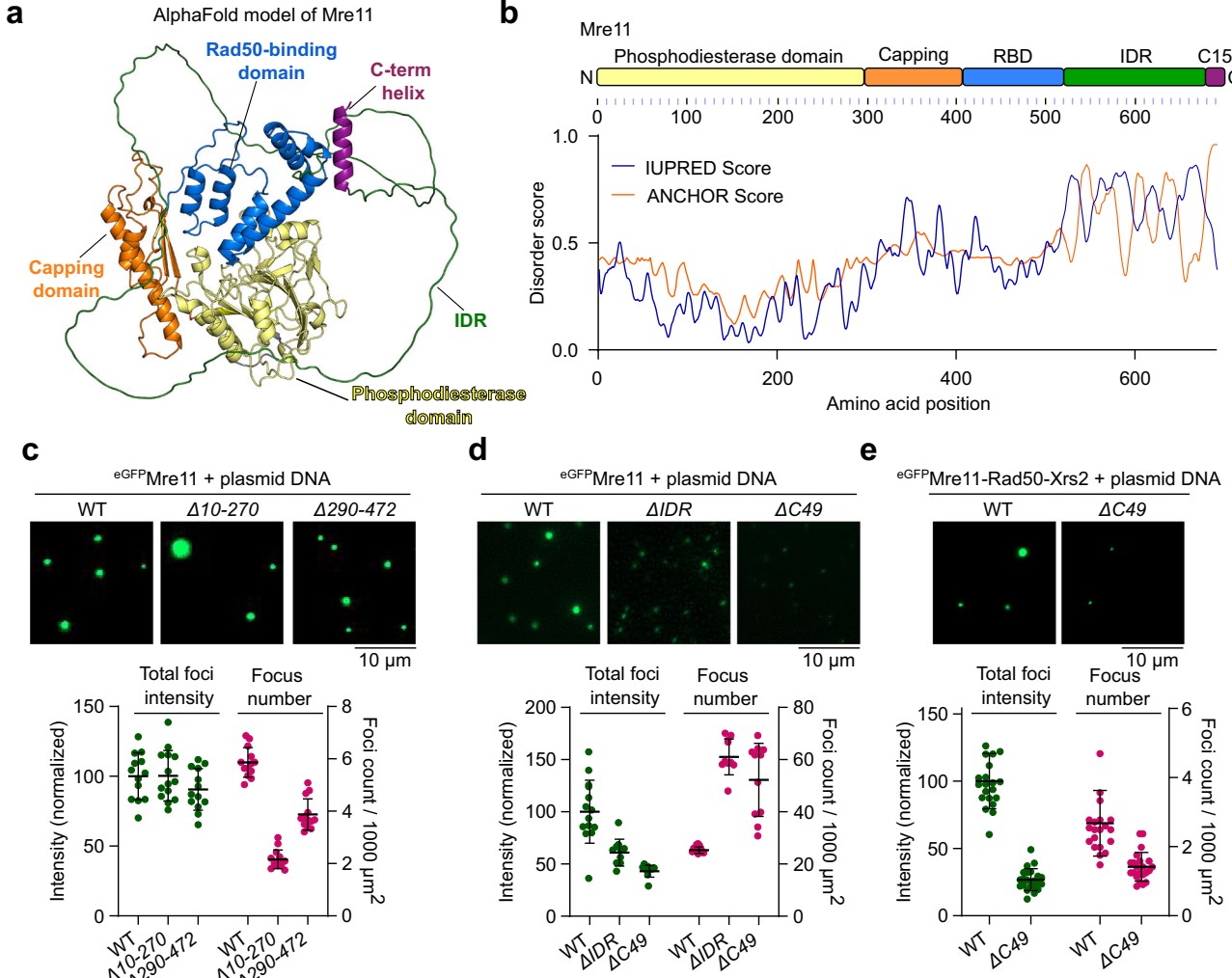

**Fig. 2 | The C-terminal IDR of Mre11 is required for condensation. a** AlphaFold2 predicted model of Mre11 (AF-P32829-F1-v4). All colored domains, excluding the C-terminus IDR and C15 helix, are in agreement with the cryoEM structure of yeast Mre11 in MR subcomplex[40]. **b** Domain structure of Mre11 and protein disorder prediction using IUPred server[79]. The ANCHOR score predicts the transition probability from a disordered to an ordered structure, dependent on a binding partner. **c**–**e** (**c**) Effect of deleting most of the phosphodiesterase domain (*Δ10−270*) (*n* = 14 fields of view), or the capping domain and Rad50-binding domain (*Δ290−472*) (*n* = 13), (**d**) deleting the disordered region (*ΔIDR*, residues 524−677)

(*n* = 10) and C-terminus (*ΔC49*, residues 644-692) (*n* = 11) on Mre11 condensation, and (**e**) deleting the C-terminus (*ΔC49*) on MRX condensation (*n* = 22). Reactions in (**c**) and (**d**) contained 200 nM Mre11. In (**c**) and (**d**), wild-type Mre11 samples have *n* = 13 and *n* = 14, respectively as controls and for (**e**), wild type MRX has *n* = 21. Quantification represents total fluorescence intensity (green) in a field of view and total number of foci per 1000 μm² (magenta). Foci intensity is normalized to the mean foci intensity of wild-type Mre11 or MRX. Error bars represent mean ± SD from the indicated (*n*) fields of view. Source data are provided as a Source data file.

of the cells also showed accumulation of fluorescent signal in discrete aggregates (Fig. 4b, e and Supplementary Fig. 4b, c). As expected, meiosis failed in both *mre11-C49* and *mre11-ΔIDR* strains, producing only dead spores (Fig. 4c).

Meiotic foci of Mre11 depend on most other DSB proteins, including RMM[41], and direct interactions have been demonstrated between Mre11, Mer2[43], and Spo11[33]. To address whether the recruitment of Mre11 during meiosis depends on RMM condensation, we quantified Mre11 foci formation in a DNA-binding defective mutant of Mer2 (*mer2-KRRR*) that compromises Mer2 focus formation in vitro and in vivo and abolishes DSB formation[9]. Similar to *mer2Δ* strains, the formation of Mre11 foci was strongly reduced in a *mer2-KRRR* mutant (Fig. 4d), despite no effect on protein levels (Supplementary Fig. 4f, g). In *mer2Δ* and *mer2-KRRR* backgrounds, Mre11 localization was analogous to the *mre11-ΔIDR* mutant, with few foci of low intensity and accumulation within a large aggregate (Fig. 4d, e).

To test whether Mer2 condensates can directly recruit Mre11 in vitro, we used fluorescently labeled Mre11 and Mer2, assembled their respective condensates separately on DNA substrates, and imaged the foci by microscopy. When Mer2 and Mre11 condensates were first assembled separately on a DNA substrate and then incubated together prior to imaging, we observed nearly absolute colocalization between the two proteins, supporting the hypothesis that Mer2 condensates recruit Mre11 through direct protein-protein interactions (Fig. 4f).

When these experiments were performed with the Mre11-ΔC49 truncation and wild-type Mer2, we observed strongly reduced colocalization as compared to wild-type Mre11 (Fig. 4f and Supplementary Fig. 4h), suggesting that the C-terminus of Mre11 is important for interaction with Mer2. While the C-terminus of Mre11 does not form condensates independently, in the presence of Mer2, it forms a halo-like structure around Mer2 condensates (Fig. 4g). This suggests that the interaction between Mer2 and the Mre11 C-terminus provides a

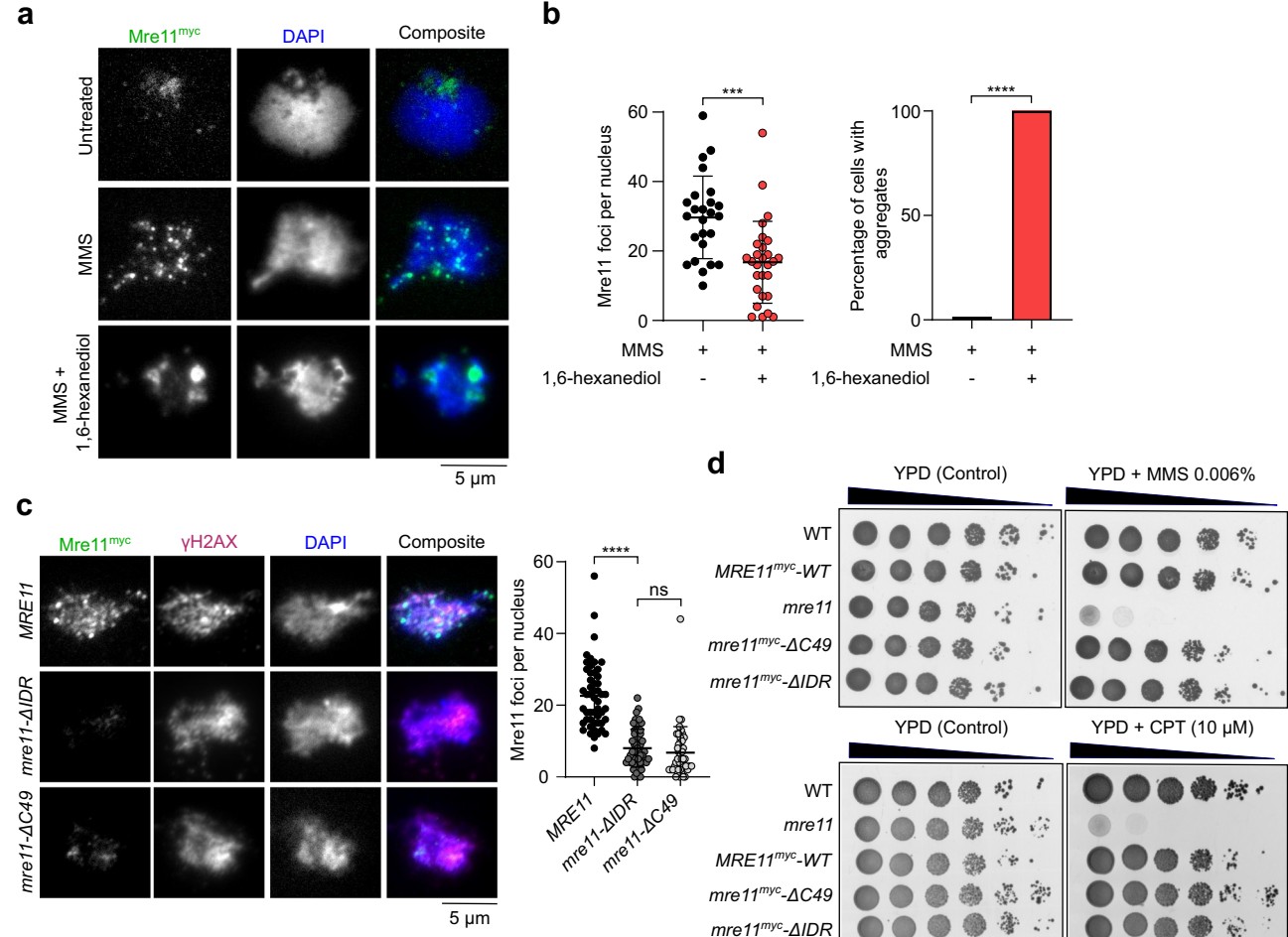

**Fig. 3 | The Mre11 IDR is required for vegetative foci formation but dispensable for DNA repair. a** Effect of 1,6-hexanediol treatment on MMS-induced Mre11myc foci visualized by immunofluorescence analysis of vegetatively growing yeast nuclear spreads. **b** Quantification of the number of Mre11 foci (left) and Mre11 aggregates (right) before and after 4-5 min of 1,6-hexanediol treatment. Error bar shows mean ± SD of $n = 25$ (untreated) and 28 (treated) cells. *** $p = 0.0002$ (two-tailed unpaired $t$ test). **c** Effect of Mre11myc truncations on MMS-induced foci formation as visualized by immunofluorescence. Error bars show mean ± SD of Mre11myc foci for *WT* ($n = 51$), *mre11-ΔIDR* ($n = 52$), and *mre11-ΔC49* ($n = 45$). **** $p < 0.0001$ (two-tailed unpaired $t$ test). **d** Sensitivity of wild-type and truncated Mre11myc strains to methylmethanesulfonate (MMS) and camptothecin (CPT). Ten-fold serial dilutions from saturated cultures are shown, with dilutions on YPD plates as a control. Source data are provided as a Source data file.

nucleation site that is sufficient for further Mre11 recruitment through self-association.

## The C-terminus of Mre11 binds a conserved motif of Mer2

To delve deeper into the interaction between Mre11 and Mer2, we used AlphaFold2[44] to model the interaction between Mre11-C49 and a Mer2 tetramer (previously defined as the Mer2 oligomeric state[9,45]). AlphaFold produced a model in which the terminal α-helix of Mre11 is positioned close to the N-terminal end of the Mer2 coiled coil (Fig. 5a). Interestingly, this region contains a conserved sequence motif (SSM1)[46] that was previously implicated in Mre11 binding[43] (Supplementary Fig. 5b). Despite the moderate confidence in the predicted interaction surface, modeling of pairs of Mer2-Mre11 homologs in different species of Saccharomycetaceae produced similar models for two out of three tested species (Supplementary Fig. 5a and Supplementary Table 6).

In the AlphaFold model, the Mre11 C-terminal helix binds anti-parallel to the Mer2 coiled coil (Fig. 5a). Mer2 residues E50, Q54, E57, and K61 (hereby referred to as EQEK) project towards the Mre11 C15 helix, and Mre11 residues L683, L686, and K690 (hereby referred to as

LLK) point towards the SSM1 region of Mer2. These residues are well-conserved across Saccharomycetaceae (Supplementary Fig. 5b, d).

To test this model, we co-expressed MBP-tagged Mre11-C49 peptide and HisSUMO-tagged Mer2 in *E. coli* and quantified protein interactions by NiNTA pulldown. While the interaction was relatively weak, anti-MBP immunoblot analysis confirmed that Mre11-C49 directly binds Mer2 (Fig. 5b). Consistent with the AlphaFold model, alanine mutations of the Mer2-EQEK or Mre11-LLK residues strongly decreased the interaction between the two partners (Fig. 5b, c). We further validated the interaction between full-length Mer2 and Mre11 using a yeast-2-hybrid (Y2H) assay and found that mutation of the Mer2-EQEK or Mre11-LLK residues abolished the interaction (Supplementary Fig. 6a). Furthermore, truncation of the Mre11 C-terminal α helix strongly reduced co-localization between eGFP-tagged Mre11 and Alexa594-labeled Mer2 condensates in vitro (Supplementary Fig. 6b-d). Hence, Mre11 binds Mer2-SSM1, and the Mre11 C-terminal 15 residues are necessary for this interaction.

In contrast to the pulldown experiments, however, the Mre11-C49 fragment was not sufficient for Mer2 binding by Y2H analysis (Supplementary Fig. 6a), suggesting that these interactions are too weak

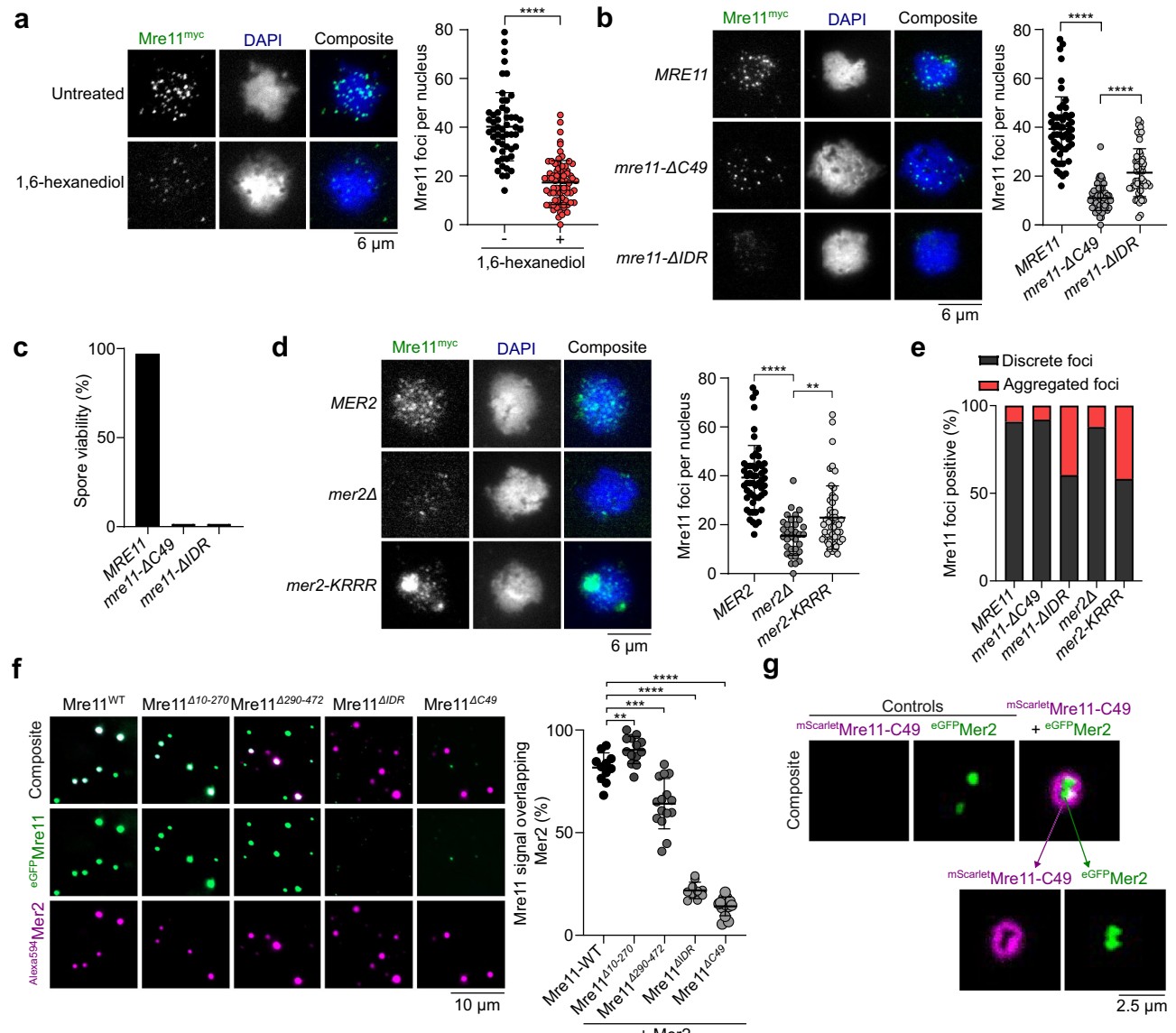

**Fig. 4 | Mer2 recruits Mre11 during meiosis. a** Effect of 1,6-hexanediol treatment on meiotic Mre11myc foci visualized by immunofluorescence analysis of yeast nuclear spreads 4 hours after transfer to SPM. Quantification shows mean and SD for untreated ($n = 53$) and 1,6-hexanediol treated cells ($n = 73$). **** $p$-value< 0.0001 (two-tailed unpaired $t$ test). **b**–**d** Immunofluorescence on meiotic nuclear spreads for myc-tagged (**b**) *MRE11-WT*, *mre11-ΔC49*, *mre11-ΔIDR* and (**d**) *MRE11-WT* in *MER2*, *mer2Δ*, and *mer2-KRRR* strains. Quantification shows mean and SD of Mre11myc foci for *WT* ($n = 55$), *mre11-ΔC49* ($n = 60$), *mre11-ΔIDR* ($n = 53$), *MER2* ($n = 55$), *mer2Δ* ($n = 33$), and *mer2-KRRR* ($n = 57$). $p$-values between Mre11-WT and *mre11-ΔC49* (<0.0001), *mre11-ΔC49* and *mre11-ΔIDR* (<0.0001), *MER2* and *mer2Δ* (<0.0001), *mer2Δ* and *mer2-KRRR* (0.0037) (two-tailed unpaired $t$ tests). **c** Spore viabilities of strains expressing wild-type or truncated Mre11myc. At least 22 tetrads were dissected for each strain ($n ≥ 88$ spores). **e** Quantification of sub-population of cells showing discrete Mre11 foci and aggregated Mre11 foci in indicated strains. Images

used for analysis were from the same experiments in (**b**) and (**d**). **f** Colocalization of fluorescently-labeled Mer2 with wild-type or truncated Mre11. Reactions containing 200 nM of Alexa594Mer2 and eGFPMre11 were assembled separately for 10 min then mixed at 1:1 ratio for 30 min prior to imaging. Quantification shows the fraction of Mer2 foci that overlap with Mre11. The number of images analyzed for each sample were WT ($n = 11$), Mre11Δ10-270 ($n = 13$), Mre11Δ290-472 ($n = 15$), Mre11ΔIDR ($n = 10$), Mre11ΔC49 ($n = 13$). Error bars represent mean ± SD from the indicated ($n$) fields of view. $p$-values compared to Mre11-WT are: Mre11Δ10-270 (0.006), Mre11Δ290-472 (0.0003), Mre11ΔIDR (<0.0001), Mre11ΔC49 (<0.0001). **g** Colocalization of fluorescently-labeled Mre11-C49 and Mer2. Condensates were assembled by mixing 800 nM of mScarletMre11-C49 and 200 nM of eGFPMer2 for 30 min prior to imaging. Controls reactions contained the same concentration but lacked either eGFPMer2 or mScarletMre11-C49, respectively. Source data are provided as a Source data file.

and are perhaps stabilized in the context of full-length Mre11 via dimerization of its N-terminal domain.

To establish the relevance of this interaction for Mre11 recruitment in vivo, we performed immunofluorescence analysis in strains that express the Mer2-EQEK mutant or Mre11-ΔC15 truncation. Consistent with our in vitro analysis, Mre11 focus number and intensity were reduced in both mutants, compared to the wild type (Fig. 5d). This was not due to reduced Mer2 or Mre11 expression, or reduced Mer2 chromatin association (Supplementary Fig. 6e–h).

To further verify the impact of the mutations on the interaction between Mer2 and Mre11 in vivo, we performed co-immunoprecipitation analyses on meiotic cell extracts in cells expressing V5-tagged Mer2 and myc-tagged Mre11. Mer2 was efficiently co-immunoprecipitated by Mre11, and mutation of the Mer2-EQEK motif or truncation of the Mre11 C-terminal α-helix reduced this interaction (Supplementary Figs. 6i, 10e).

To address whether the interaction between Mer2 and Mre11 is important for their meiotic function, we analyzed DSB formation at

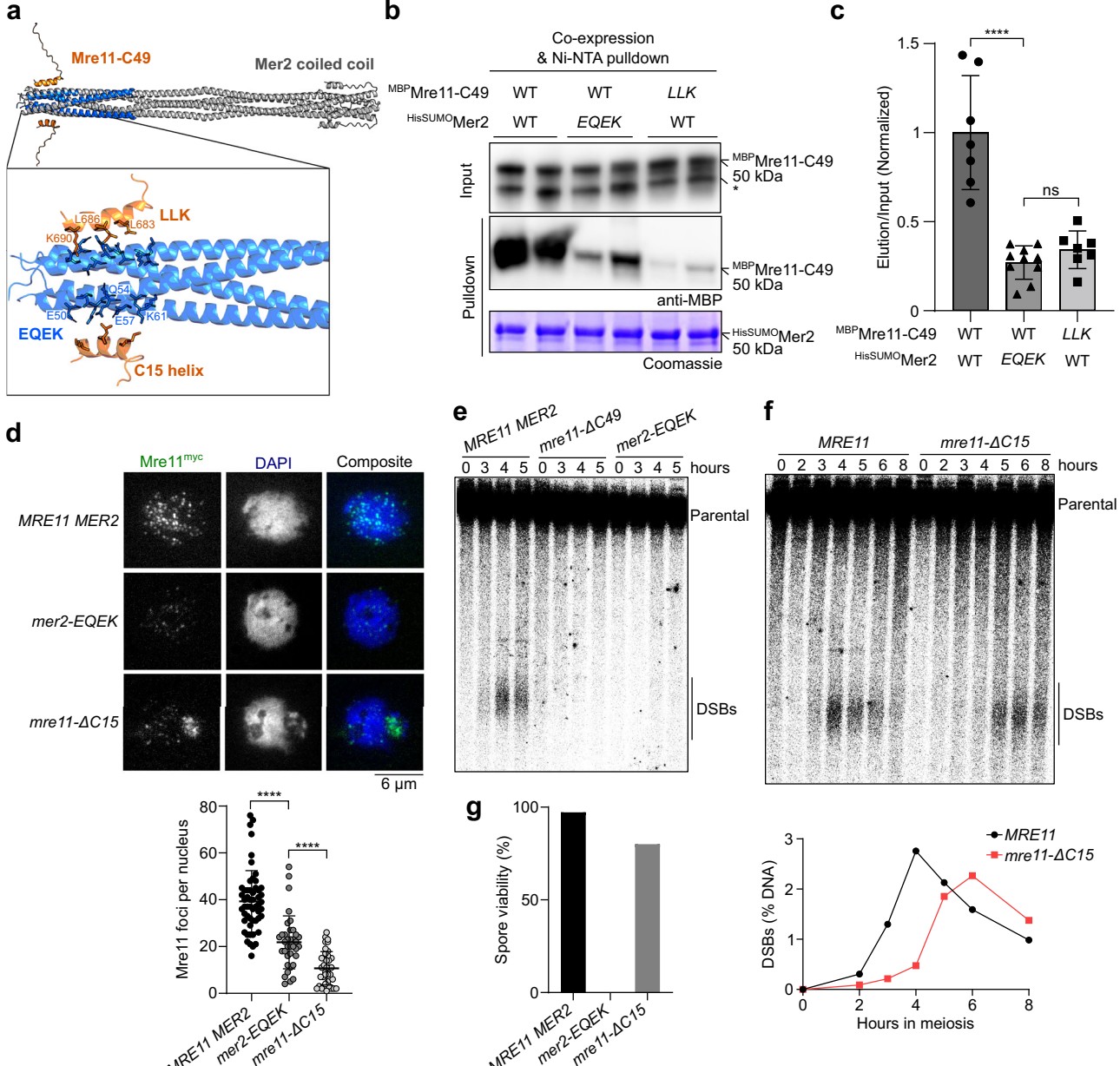

**Fig. 5 | The C-terminus of Mre11 binds a conserved motif of Mer2. a** AlphaFold2 predicted model of Mre11 C-terminal residues (644–692, Mre11-C49) (orange) and Mer2 (41-100) (blue), aligned to the Mer2 coiled-coil domain (residues 41–247) (gray). Magnified-view shows amino acid residues at the predicted interaction surface, Mre11 residues L683, L686, and K690 (LLK) and Mer2 residues E50, Q54, E57, and K61 (EQEK). **b** Co-expression based pulldown assay between wild type and mutant $^{MBP}$Mre11-C49 (prey) and $^{HisSUMO}$Mer2 (bait). The LLK and EQEK mutants have the respective residues mutated to alanine. Input and pulldown immunoblots are visualized with anti-MBP primary antibody. Asterisk (*) denotes free MBP. **c** Quantification of anti-MBP elution/input signal from pulldown in panel (**b**),

normalized to the wild type. Error bars represent mean ± SD from independent samples from $n = 7$ (wild type), $n = 10$ (Mer2-EQEK), and $n = 7$ (Mre11-LLK) cultures. $p$-values between WT and *mer2-EQEK* ($< 0.0001$) and *mer2-EQEK* and *mre11-LLK* (0.15) (two-tailed unpaired $t$ tests). **d** Immunofluorescence on meiotic nuclear spreads of Mre11$^{myc}$ in WT, *mer2-EQEK*, and *mre11-ΔC15* strains. Quantification of Mre11$^{myc}$ foci in WT ($n = 54$), *mer2-EQEK* ($n = 36$), and *mre11-ΔC15* ($n = 38$). Error bars represent mean ± SD. **** $p < 0.0001$ (two-tailed unpaired $t$ tests). **e, f** Southern blot analysis of meiotic DSB formation at the *GAT1* hotspot. Quantification of panel F show mean from $n = 2$ experiments. **g** Spore viabilities of wild-type ($n = 144$) and mutant strains ($n = 80$ spores). Source data are provided as a Source data file.

the *GAT1* hotspot by Southern blotting. Similar to the *mre11-ΔC49* mutant, little or no DSB signal was detected in *mer2-EQEK* cells (Fig. 5e), while the *mre11-ΔC15* mutation caused reduced and delayed DSB formation (Fig. 5f), even though the progression of meiosis was unchanged (Supplementary Fig. 6j). Consistently, *mer2-EQEK* mutant spores were completely inviable while the *mre11-ΔC15* truncation showed 80% spore viability (Fig. 5g). Since *mre11-ΔC15* does not phenocopy the *mre11-ΔC49* and *mer2-EQEK* mutations, we conclude

that the Mer2 SSM1 motif and the C-terminus of Mre11 exert other meiotic functions than the direct Mer2-Mre11 interaction identified here.

## The Mre11 C-terminus contains a SUMO-interaction motif

Several DSB proteins, including Mer2, were shown to be SUMOylated during meiosis, and SUMOylation regulates all aspects of meiotic prophase I, including DSB formation[47]. Notably, SUMOylation sites

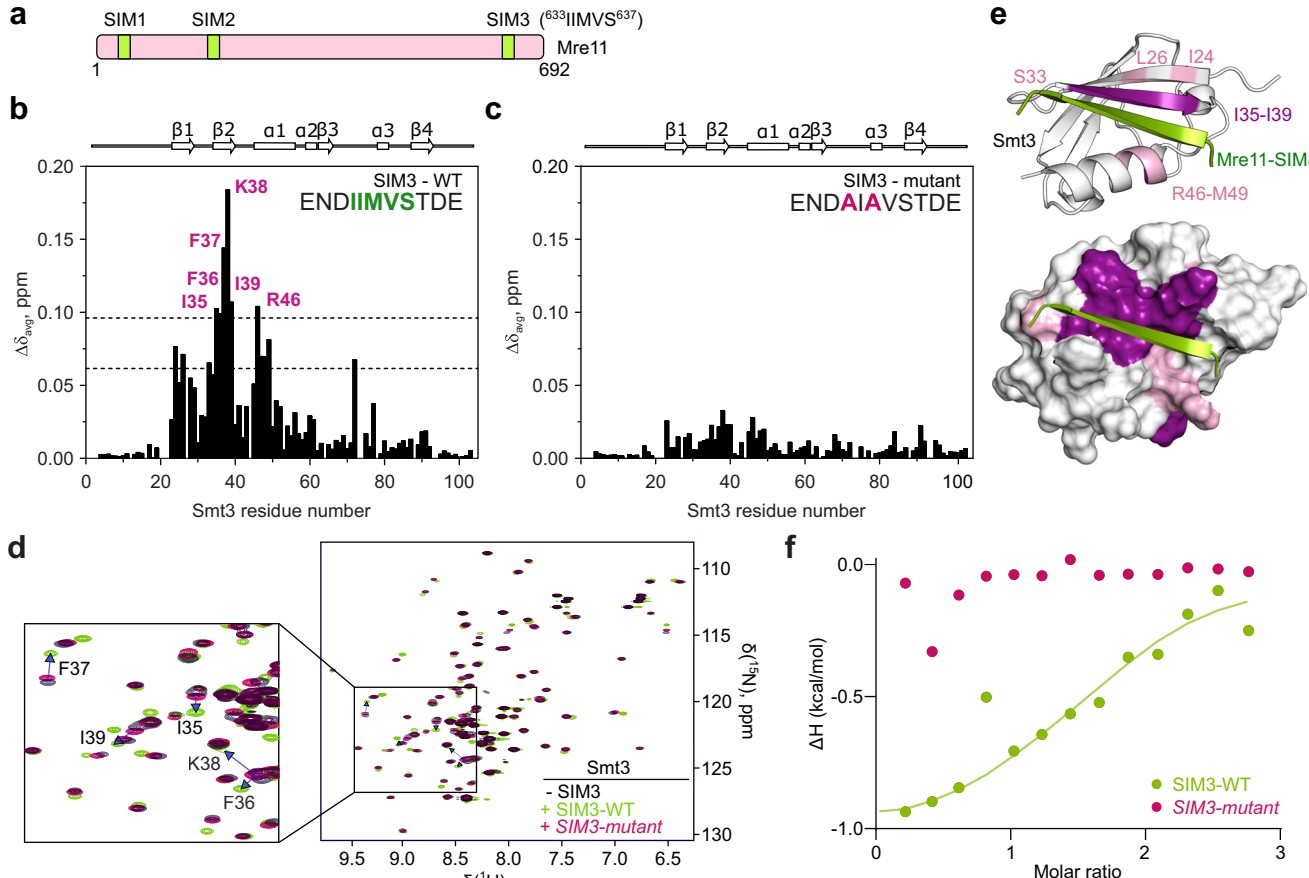

**Fig. 6 | The Mre11 C-terminus contains a SUMO-interaction motif. a** Schematic representation of SUMO-interacting motifs (SIMs) in Mre11. **b, c** Average backbone amide chemical shift perturbations ($\Delta\delta_{avg}$) of Smt3 in the presence of 1.2 molar equivalents of (**b**) wild-type or (**C**) mutant SIM3 peptide (sequences indicated). The secondary structure of Smt3 is shown above the plots. The horizontal lines in (**b**) correspond to the average $\Delta\delta_{avg}$ (avg) plus one or two standard deviations (stdev). **d** Overlaid [¹H,¹⁵N] HSQC spectra of Smt3 in its free form (black) or with 1.2 molar equivalents of wild-type (green) or mutant (pink) SIM3 peptides. Blue arrows indicate residues that experience the highest chemical shift. **e** Chemical shift

mapping of the wild-type SIM3 peptide binding. Smt3 is shown as (top) cartoon and (bottom) molecular surface colored by the $\Delta\delta_{avg}$ values (pink: $\Delta\delta_{avg}$ > avg + stdev; violet: $\Delta\delta_{avg}$ > avg + 2*stdev). The bound SIM3 peptide is in green. The disordered regions at Smt3 N- and C-termini are omitted for clarity. **f** Normalized heat per peak $\Delta H$ (kcal/mol) as a function of molar ratio (peptide/protein concentration) measured by isothermal titration calorimetry upon titration of wild-type (green) or mutant (pink) SIM3 peptide on Smt3. The solid line shows the best fit to a single binding site model for the wild-type peptide. Source data are provided as a Source data file.

were mapped in the SSM1 region of Mer2 that also harbors the EQEK residues.

Mre11 has two previously identified SUMO-interacting motifs located in its phosphodiesterase domain (SIM1 and SIM2) that have been implicated in DSB repair in both mitotic and meiotic cells[48]. Using the GPS-SUMO tool[49], we identified a third potential SIM, here called SIM3 (IIMVS), located towards the end of the Mre11 IDR (Fig. 6a and Supplementary Fig. 7a). AlphaFold prediction yielded a high-confidence model in which Mre11-SIM3, which is otherwise disordered, assumes a β-sheet structure in the presence of Smt3, as expected from a bona fide SIM[50] (Supplementary Fig. 7b–d).

To validate the Mre11-SIM3, we performed Nuclear Magnetic Resonance (NMR) spectroscopy on purified and isotopically labeled U-[¹³C, ¹⁵N] yeast Smt3 in the presence of a synthetic Mre11-SIM3 peptide (Fig. 6b and Supplementary Fig. 7e). As a control, we mutated residues corresponding to Mre11 I633 and M635 to alanine, predicted to contribute to the interaction with Smt3 (Supplementary Fig. 7b). NMR spectroscopy analysis indicated that the wild-type Mre11-SIM3 peptide induced strong backbone amide chemical shift to Smt3 residues I35, F36, F37, K38, I39 and R46, in contrast to the mutant peptide (Figs. 6b–d). Residues I35–I39 form part of the second β-sheet of Smt3 and R46 is located on the first α-helix that are both predicted to bind

the SIM3 peptide (Fig. 6e). In addition, analysis of methyl binding shifts closely agrees with the backbone amide chemical shift perturbation and further validates the AlphaFold-predicted model between Mre11-SIM3 and Smt3 (Supplementary Fig. 8).

To evaluate the affinity of Smt3 for Mre11-SIM3, we performed isothermal titration calorimetry (ITC) binding experiments between Smt3 and the Mre11-SIM3 peptides. ITC analysis confirmed that the wild-type SIM3 peptide binds Smt3 with a $K_D$ of 20 ± 10 μM, typical for SUMO-SIM interactions[51,52]. In contrast, the mutant SIM3 peptide failed to interact with Smt3 (Fig. 6f and Supplementary Fig. 9).

## SIM3 contributes to Mre11 recruitment and meiotic DSB formation

To test the role of Mre11-SIM3 in vivo, the core SIM3 motif (IIMVS) was substituted with alanines. *mre11-SIM3* mutant cells showed a small reduction in meiotic Mre11 foci (Fig. 7a), which was associated with delayed DSB formation (Fig. 7b and Supplementary Fig. 10a). Combining the *SIM3* mutation with truncation of the Mre11 C-terminal α-helix (*mre11-SIM3+ΔC15*) further reduced Mre11 foci and abolished DSB formation (Fig. 7a, b), while protein levels remained unchanged (Supplementary Fig. 10b, c). Consequently, the *mre11-SIM3* mutant had slightly reduced spore viability (86%), while *mre11-SIM3+ΔC15* spores

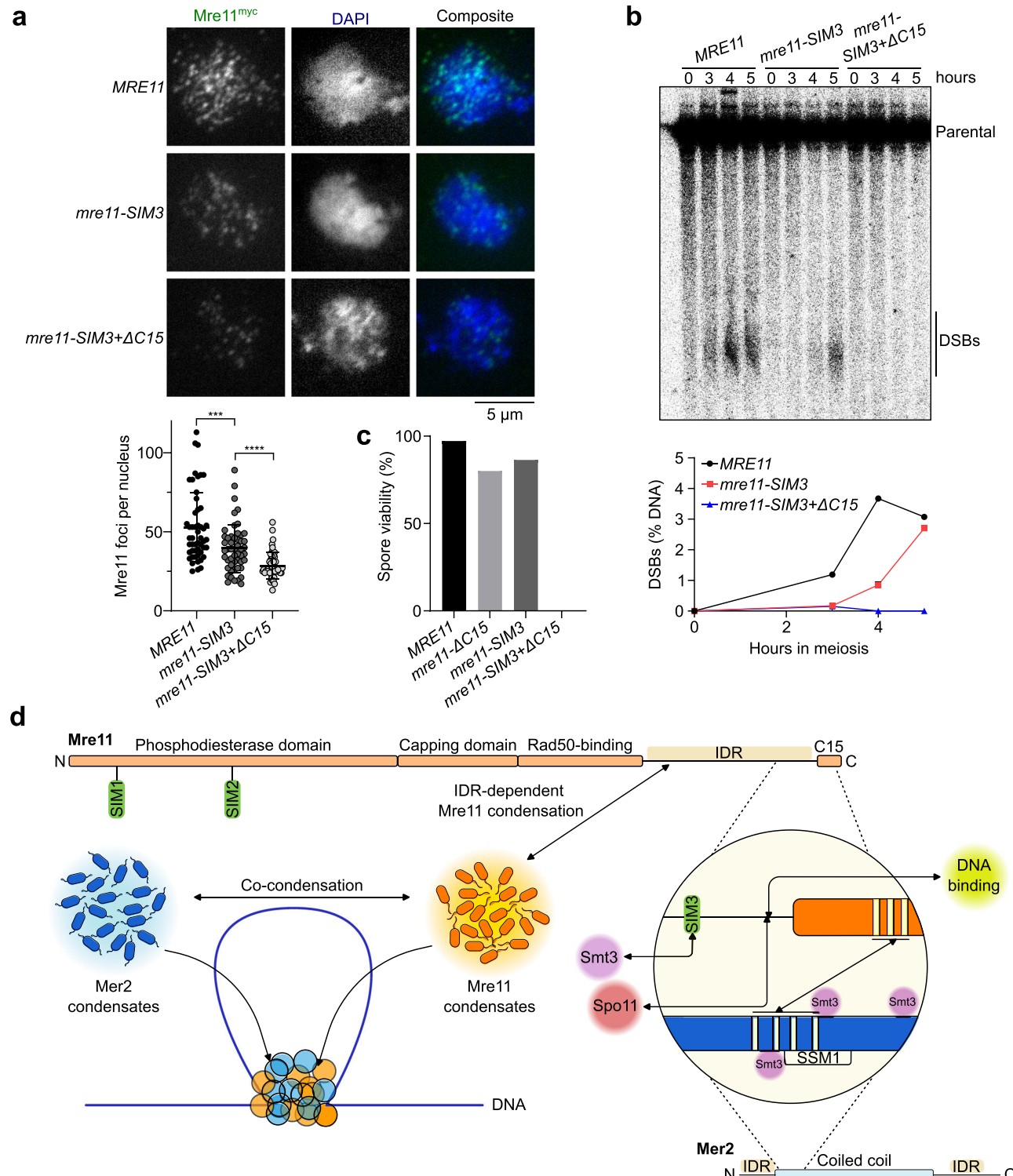

**Fig. 7 | A SUMO-SIM interaction fosters the recruitment of Mre11 during meiosis. a** Immunofluorescence on meiotic nuclear spreads of myc-tagged *MRE11-WT*, *mre11-SIM3*, and *mre11-SIM3+ΔC15* cells. Quantification of Mre11^myc foci show mean ± SD of *n* = 50 cells per strain. *p*-values between *MRE11-WT* and *mre11-SIM3* (0.0006) and *mre11-SIM3* and *mre11-SIM3+ΔC15* (< 0.0001) (two-tailed unpaired t-tests). **b** Southern blot analysis of meiotic DSB formation at the *GAT1* hotspot. Quantifications show the mean of *n* = 2 independent experiments. **c** Spore viabilities of wild-type and mutant *MRE11* strains (*n* = 144, WT; *n* = 80, *ΔC15*; *n* = 88, *SIM*; *n* = 80, *SIM3+ΔC15*). **d** Schematic model of the mechanisms that drive Mre11 recruitment during meiosis. Mre11 undergoes DNA-dependent condensation, driven by its IDR, and can mingle with Mer2 to form joint condensates. Mre11 is recruited to the DSB machinery via its C-terminus through a SUMO-SIM interaction and direct binding with Spo11 and Mer2. Source data are provided as a Source data file.

were completely inviable (Fig. 7c). The *mre11-SIM3* mutant also showed a minor delay in meiotic progression, while progression was accelerated in the *mre11-SIM3+ΔC15* mutant compared to the wild-type, likely due to the absence of DNA breaks (Supplementary Fig. 10d). We conclude that SIM3 promotes the recruitment of Mre11 to the meiotic DSB machinery.

To test the hypothesis that SIM3 contributes to the interaction between Mre11 and Mer2, we analyzed the Mre11-Mer2 interaction by Y2H. While truncation of the terminal 15 amino acids of Mre11 abolished the interaction with Mer2, mutation of the SIM3 motif had no discernable impact (Supplementary Fig. 10e). We note, however, that Mer2 may not be SUMOylated in this system, which employs mitotically cycling cells. To overcome this issue, we analyzed Mre11-Mer2 interaction by co-immunoprecipitation in meiotic cells. Here, truncation of the Mre11 C-terminal 15 residues or mutation the SIM3 motif both reduced the interaction with Mer2, and combining the mutations abolished the interaction completely (Supplementary Fig. 10f), while showing no phenotype in vegetative conditions (Supplementary Fig. 10g, h). Hence, this demonstrates that the Mre11 C-terminal helix and SIM3 collaborate to promote Mre11 recruitment during meiosis. Nevertheless, the potential contribution of Mer2 SUMOylation to productive interaction with Mre11 remains unclear.

## Discussion

The in vitro condensation activity of Mre11 and MRX is reminiscent of that of Rec114-Mei4 and Mer2 that also form dynamic and reversible macromolecular assemblies in the presence of DNA[9]. DNA promotes Mre11 condensation and likely acts as a scaffold that then recruits free soluble protein through homotypic self-association. Similar to RMM, Mre11 and MRX condensates likely involve electrostatic interactions between positively charged residues and the DNA backbone. Mre11 condensates are also stabilized by divalent metal ions that presumably inhibit intramolecular repulsion within the DNA substrate, and are sensitive to 1,6-hexanediol, indicating that they depend on weak hydrophobic interactions.

While both Mre11 and MRX condensates share similar properties, they don't fully mirror each other in their behavior, suggesting that the presence of Rad50 and Xrs2 impacts MRX condensates, consistent with a previous observation that Rad50 self-interaction drives Mre11-Rad50 oligomerization and is required for proper DNA-binding and Mre11 endonuclease activity[36].

The spontaneous assembly of Mer2, Rec114-Mei4, and Mre11/MRX condensates in vitro contrasts with the more stringent assembly of their respective foci in vivo. Indeed, Mer2 foci depend on the axis protein Hop1, Rec114 foci depend on Mer2, and Mre11 foci depend on Mer2 and likely most other DSB proteins[16,41,53–55]. These observations support a hierarchical mechanism of recruitment of the DSB proteins, in which Hop1 recruits and may promote the local enrichment of Mer2, which in turn recruits Rec114-Mei4 and facilitates accumulation, followed by the recruitment of the Spo11 complex[9]. Mer2 also recruits Mre11 and may promote its local accumulation through multivalent interactions that could be further cooperatively stabilized by binding to SUMOylated targets, including perhaps Mer2 itself (Fig. 7d).

The Mre11 C-terminal IDR is necessary, but not sufficient, for condensation. The N-terminal region is also necessary, and likely confers two properties[56]: a DNA-binding activity through the nuclease domain, which presumably provides anchor points, and dimerization, which presumably contributes to multivalency.

Although the Mre11 C-terminus is not sufficient for condensation, the presence of Mer2 condensates triggers the accumulation of Mre11-C49 peptides. This shows that Mer2 locally concentrates Mre11-C49, allowing it to reach a threshold required for self-association. Rather than mingling as joint condensates like the full-length proteins do, Mre11-C49 peptides form a shell around Mer2 condensates. Mre11-C49 peptides presumably do not penetrate Mer2 condensates because

their homotypic interactions are favored over heterotypic interactions with Mer2.

Like Mre11 condensates in vitro, Mre11 foci in vivo are sensitive to 1,6-hexanediol. In vegetative conditions, hexanediol treatment led to hyper-accumulation of Mre11 into one or a few large aggregates. Similar aggregates were also observed in a sub-population of cells harboring *mre11-ΔIDR* or *mer2-KRRR* mutations. The basis of this hyperaccumulation remains unclear. It may reflect the intrinsic self-assembly property of Mre11 or the MRX complex, analogous to synaptonemal complex proteins that form polycomplexes under pathological conditions[57,58], or may reflect a binding activity of Mre11/MRX to an unknown cellular factor. One way to understand this is that, under normal (vegetative or meiotic) conditions, the affinity of Mre11 to its substrate and cellular partners allows its localization throughout the nucleus. In some pathological conditions (e.g., upon hexanediol treatment in vegetative cells, or in a *mer2-KRRR* background in meiotic cells), the network of interactions that fine-tunes Mre11 localization is compromised, leading to its local hyperaccumulation (by self-assembly or by tethering to unknown cellular factors).

While our work indicates that biomolecular condensation is an intrinsic biochemical property of Mre11 and is consistent with a role of self-assembly during meiosis, it also suggests that IDR-mediated condensation is dispensable for the repair of exogenous DSBs in vegetative contexts. MMS-induced focal accumulation of Mre11 may therefore reflect a side effect of the self-assembly property of Mre11 that is relevant to other contexts, for instance, meiosis. Alternatively, it is also possible that IDR-dependent focal accumulation could have functional consequences in DSB repair that did not manifest in our survival-based assay. In addition, beyond exogenous DNA damage repair and meiotic DSB formation, Mre11 also has established roles in other cellular contexts, including checkpoint activation, telomere maintenance, and stabilization of stalled replication forks[28–30]. Mre11 self-assembly behavior could potentially influence any of these functions.

Besides exploring the condensation properties of Mre11, we also investigated how Mre11 is recruited during meiosis. It had previously been shown that the C-terminus of Mre11 is essential for the formation of meiotic DSBs, but dispensable for DSB repair in mitotically cycling cells[31,41]. Here, we show that this essential meiotic function of the Mre11 C-terminus involves multiple mechanisms that collaborate to promote Mre11 recruitment to recombination sites (Fig. 7d).

First, we show that the C-terminal α-helix of Mre11 directly binds the Mer2 SSM1 motif located on the N-terminal side of the tetrameric coiled coil. AlphaFold modeling and mutagenesis identified Mre11-LLK and Mer2-EQEK residues as being responsible for this interaction. Truncating the Mre11 terminal α-helix or mutating the Mer2-EQEK residues in vivo confirmed the functional importance of this interaction for Mre11 foci formation and DSB formation. These data are in line with previous yeast-two-hybrid and pulldown experiments that revealed an interaction between *S. cerevisiae* Mre11[43] and Mer2 and the *A. thaliana* orthologs MRE11 and PRD3[59]. We note, however, that the Mer2 alleles previously reported to reduce Mre11 interaction involved amino acids predicted to point inside the Mer2 tetrameric coiled coil, which are therefore likely to affect Mre11 interaction indirectly by compromising the structural stability of Mer2 (Supplementary Fig. 5c).

Second, we identified and biochemically validated a previously uncharacterized SUMO-interacting motif located within the Mre11 IDR and showed that this motif also participates in Mre11 recruitment and DSB formation during meiosis. While the meiotic phenotypes of the *mre11-SIM3* mutant were relatively modest, we find that SIM3 collaborates with the C-terminal α-helix of Mre11 for productive recruitment to precursor DSB sites.

Third, a recent study demonstrated that the Mre11 C-terminus is also important for direct interaction with Spo11[33]. While the binding

sites were not precisely identified, this interaction implicates Mre11 residues 663 to 676, just upstream of the terminal α-helix. It was noted that a mutant lacking the last 16 amino acids of Mre11 was defective in meiosis despite no effect on Spo11 binding, indicating that interaction with Spo11 is not sufficient for functional recruitment of Mre11. Our data explains this result by demonstrating that the Mre11 terminal α-helix binds to Mer2.

Finally, as discussed above, the C-terminal region of Mre11 also contributes to its higher-order assembly, suggesting that it may constitute yet another meiosis-specific function of the Mre11 C-terminal tail. Nevertheless, in the absence of a separation-of-function mutant that specifically abolishes this activity, its functional importance for meiotic DSB formation remains uncertain. In summary, these findings indicate that the recruitment of Mre11 during meiosis involves at least three, perhaps four, independent functions of the Mre11 C-terminal tail.

The MRX complex plays key functions in the maintenance of genomic integrity throughout eukaryotes[60,61]. However, its role in promoting the formation of Spo11-dependent DSBs during meiosis has only been reported in budding yeast and *C. elegans*[17–19,62]. Indeed, MRX orthologs are not required for meiotic DSB formation in *A. thaliana*[63], *S. pombe*[64], and mice[65,66]. In *C. elegans*, MRE-11 and RAD-50 are necessary for DSB formation[62,67], but the ortholog of Xrs2, NBS-1, is not[68].

It has been suggested that the requirement for the MRX complex prior to DSB formation serves to coordinate DSB formation with downstream repair, thereby minimizing genomic instability[41]. Supporting this idea, DSBs detected in wild-type cells are typically fully resected, indicating that processing occurs more rapidly than break accumulation. This observation is consistent with tight coordination between DSB formation and repair[69–72].

In *S. cerevisiae*, the binding of Mre11 to DSB hotspots depends on the presence of all other DSB proteins except Rad50[41]. This suggests that the MRX complex may be the final component recruited to the DSB machinery and raises the possibility that MRX recruitment may trigger Spo11's catalytic activity, though the underlying mechanism remains unknown.

If the recruitment of Mre11 constitutes the final step in the assembly of the DSB machinery prior to triggering Spo11-dependent cleavage, SUMOylation of the DSB machinery could serve to mark licensed pre-DSB complexes and/or allow for reversible interactions.

What are the SUMOylated targets bound by Mre11 prior to DSB formation? SUMOylation was previously shown to regulate the key events of meiotic prophase I, including DSB formation, and thousands of SUMOylation sites were mapped on meiotic proteins[47]. Amongst those, thirteen sites were identified within Mer2, including several close to the SSM1 motif. Given the physical proximity of the Mre11$^{LLK}$ and Mer2$^{EQEK}$ interaction regions and Mre11-SIM3, it is tempting to speculate that SIM3 might be interacting with SUMOylated Mer2 during meiosis, which would presumably serve as an anchor to stabilize this interaction. Indeed, this is supported by co-immunoprecipitation experiments, which show that mutating SIM3 partly affects interaction with Mer2. However, the recruitment of Mre11 through the SIM3 motif may involve other SUMOylated proteins involved in DSB formation, including Spp1, Rec114, Hop1, Red1, and cohesin[47].

## Methods

### Preparation of expression vectors
Oligonucleotides (oligos) used in this study were purchased from Sigma-Aldrich. The sequences of the oligos used are listed in Supplementary Table 1. Plasmids generated in this study were verified by sequencing and are listed in Supplementary Table 2. Peptides were ordered from GenScript or synthesized in the Ballet laboratory and are listed in Supplementary Table 3.

The expression vector for Mre11$^{10xHis}$ was produced by PCR amplification of the *MRE11* gene from yeast genomic DNA (SK1 strain) using primers cb1351 and cb1352 and Gibson assembly into a BamHI and EcoRI digestion fragment of pFastBac1 to yield pCCB865. The sequence coding for eGFP was cloned into the BamHI site of pCCB865 to produce the expression vector for $^{eGFP}$Mre11, pCCB942. The expression vector for Rad50 was produced by PCR amplification of the *RAD50* gene from SK1 genomic DNA using primers cb1353 and cb1354 and Gibson assembly into a BamHI and EcoRI digestion fragment of pFastBac1 to yield pCCB866. The expression vector for Xrs2$^{2xFlag}$ was produced by PCR amplification of the *XRS2* gene from SK1 genomic DNA using primers cb1355 and cb1356 and Gibson assembly into a BamHI and EcoRI digestion fragment of pFastBac1 to yield pCCB867.

Expression vectors for $^{eGFP}$Mre11$^{10xHis}$-*Δ10-270* (pry7), $^{eGFP}$Mre1$^{10xHis}$-*Δ290-472* (pry5), $^{eGFP}$Mre1$^{10xHis}$-Δ*IDR* (pry6), $^{eGFP}$Mre1$^{10xHis}$-Δ*C49* (pCCB943), and $^{eGFP}$Mre1$^{10xHis}$-Δ*C15* (pry44) were amplified by inverse PCR from pCCB942 using primers pp26 and pp27, pp28 and pp29, pp24 and pp25, cb1435 and cb1458, pp73 and pp74 respectively. The amplified product was gel extracted, phosphorylated, and ligated to generate the truncations. The expression vector for $^{eGFP}$Mre11-IDR (pry41) was generated by Gibson assembly using a backbone amplified from pCCB942 with primers pp55 and pp61 and the Mre11-IDR sequence amplified from pCCB942 using primers pp59 and pp60.

The vector for expression of $^{MBP}$Mre11-C49 and $^{HisSUMO}$Mer2 (pCCB1040) was based on a pET-Duet1 vector with the sequence coding for $^{MBP}$Mre11-C49 cloned within the first position (SacI site) and the sequence coding for $^{HisSUMO}$Mer2 cloned at the second position (XhoI site). The EQEK mutations and LLK mutations were introduced by PCR amplification of pCCB1040 with primers cb1561 and cb1562, and pp84 and pp85, followed by phosphorylation and self-ligation to yield plasmids pry59 and pry61, respectively.

The expression vector for $^{mScarlet}$Mre11-C49 (pry109) was generated by performing a three-fragment Gibson assembly of a pET28a backbone containing MBP, followed by a TEV cleavage site amplified from pCCB785 using primers cb1486 and pp120, Mre11-C49 amplified from pCCB1040 using primers pp92 and pp136, and mScarlet amplified from pCCB785 using primers pp148 and pp149. Similarly, $^{MBP}$Mre11-C49 (pry99) was generated by two-fragment Gibson assembly of the fragment generated from amplification of pCCB785 using primers cb1486 and pp120 and Mre11-C49 amplified from pCCB1040 using primers pp136 and cb1559. Expression vectors for $^{Alexa594}$Mer2 (pCCB750), $^{HisSUMO}$Mer2 coiled coil (pDD065) and $^{eGFP}$Mer2 (pCCB777) were previously described[9,45]. The MBP expression construct was generated by Gibson assembly of a pET28a backbone and MBP amplified from pry99 using primers pp127 and pp120 and pp135 and pp132, respectively.

The pET28b expression vector for Smt3 (pCCB998) was a kind gift from Chris Lima[73]. Plasmids for yeast 2-hybrid, pWL1592 (pGBDU-C1-Mer2), pWL1596 (pGAD-C1-Mre11), and pWL1565 (pGAD-C1) were generously provided by John Weir. Mutations in the coding region of Mre11 were introduced in pWL1596 via PCR mutagenesis using the primers RB70 and RB267 to generate pGAD-C1-Mre11-*SIM3* (pNH1371), RB268 and RB269 to generate pGAD-C1-Mre11-Δ*C15* (pNH1372), RB313 and RB308 to generate pGAD-C1-Mre11-LLK (pNH1423), RB309 and RB310 to generate pGAD-C1-Mre11-C49 (pNH1424). EQEK mutations were introduced into the coding region of Mer2 in pWL1592 using RB311 and RB312 to generate pGBDU-C1-Mer2-EQEK (pNH1422).

### Expression and purification of recombinant proteins
Recombinant baculoviruses were produced by Bac-to-Bac Baculovirus Expression System (Invitrogen) following the manufacturer's instructions. For every induction, 1 L culture containing $2 \times 10^6$ *Spodoptera frugiperda* (Sf9) cells/ml were infected with a Multiplicity of Infection (MOI) of 2.5 for each of the viruses. Viruses generated from pCCB865, pCCB866, and pCCB867 were used for the expression of Mre11$^{10xHis}$-

Rad50-Xrs2[2xFLAG] (MRX) and pCCB942, pCCB866, and pCCB867 were used for the expression of [eGFP]Mre11[10xHis]-Rad50-Xrs2[2xFLAG] ([eGFP]MRX). Full-length Mre11[10xHis] and [eGFP]Mre11[10xHis] were expressed using viruses generated from pCCB856 and pCCB942, respectively. Mre11 truncations [eGFP]Mre11[10xHis]-Δ10-270, [eGFP]Mre11[10xHis]-Δ290-472, [eGFP]Mre11[10xHis]-ΔIDR, and [eGFP]Mre11[10xHis]-ΔC49 were expressed using viruses generated from pry7, pry5, pry6, and pCCB943, respectively.

Prior to harvest, Sf9 cells were allowed to infect for 62 h at 27 °C at 80 rpm, following which cells were pelleted at 500 rcf, washed once with 1x PBS, snap frozen in liquid nitrogen, and stored at − 80 °C, or used for purification. All subsequent purification steps were carried out at 0–4 °C.

The following protocol was used for the purification of His-tagged [eGFP]Mre11 and [eGFP]Mre11 truncations: Frozen pellets were resuspended in lysis buffer (25 mM HEPES, pH 7.5, 20 mM imidazole, 0.1 mM DTT, Roche Complete Tablet (11836170001), and 0.3 mM PMSF) and made up to a total volume of 35 ml. The samples were transferred to a beaker and osmotic lysis was performed by slowly adding 5 ml of 5 M NaCl (final 500 mM) and 10 ml of 50% (vol/vol) glycerol (final 10%) while gradually mixing with a stir bar for 30–40 mins. Lysed cells were centrifuged at 30,000 rpm for 30 mins and soluble fraction was used for affinity chromatography. 1 ml Ni-NTA resin (Thermo Scientific, 88223) was pre-equilibrated with wash buffer (25 mM HEPES, pH 7.5, 500 mM NaCl, 10% glycerol, 20 mM imidazole, 0.1 mM DTT, 0.3 mM PMSF) and batch incubated with the soluble fraction for 1 h. The resin was washed extensively in wash buffer and eluted with wash buffer containing 500 mM imidazole. Peak fractions were pooled and loaded onto a Superdex 200 Increase 10/300 GL column pre-equilibrated with SEC buffer (25 mM HEPES, 10% glycerol, 1 mM DTT, 2 mM EDTA, 300 mM NaCl). Following size-exclusion chromatography, fractions containing protein were concentrated using a 30 kDa MWCO Amicon ultra centrifugal filters (Millipore), aliquoted, snap froze in liquid nitrogen, and stored at − 80 °C.

For fluorescence labeling of Mre11, the Ni-NTA eluate was dialyzed several times to remove traces of imidazole. Labeling reaction was performed using Alexa Fluor 488 Protein Labeling Kit (Invitrogen, A10235) that has a succinimidyl ester moiety that reacts with primary amines. After 1 h conjugation at room temperature, the unbound fluorophore was removed by size-exclusion chromatography as described above.

Purification of recombinant His- and Flag-tagged [eGFP]MRX complexes and truncations were performed essentially as described[74]. Briefly, following osmotic lysis, the soluble extract was used for sequential affinity chromatography with Ni-NTA resin (Thermo Scientific, 88223) and anti-FLAG M2 affinity gel (Sigma, A2220). Peak eluted fractions were pooled, aliquoted, and snap frozen.

For the expression of recombinant [mScarlet]Mre11-C49, [MBP]Mre11-C49, and MBP in E. coli, respective plasmids, i.e., pry109, pry99, and pry94, were transformed in BL21 cells and plated on LB plates containing kanamycin. Cells were then cultured in LB media at 37 °C to an optical density (OD$_{600}$) of 0.6. Expression was carried out for 20 h at 16 °C with 0.3 mM isopropyl β-D-1-thiogalactopyranoside (IPTG) for pry109 and pry99 and 3 hours at 37 °C with 1 mM IPTG for pry94, following which cells were pelleted at 3000 rcf, washed once with 1x PBS, snap frozen in liquid nitrogen, and stored at − 80 °C, or used for purification. Cell pellet was resuspended in lysis buffer containing 25 mM HEPES, 500 mM NaCl, 0.1 mM DTT, 0.01% NP40, 10% glycerol, 1 mM PMSF, Protease Inhibitor Cocktail (PIC, 1:100, Sigma), and 2 mM EDTA. Cells were lysed by sonication (15 W, 5 mins, 5 sec pulse) and centrifuged at 20,000 rpm for 20 mins. Soluble extract was incubated for 1 h with 1.5 ml amylose resin (E8021L, NEB), pre-equilibrated with wash buffer (25 mM HEPES, 500 mM NaCl, 0.1 mM DTT, 0.01% NP40, 10% glycerol, 0.5 mM PMSF, Protease Inhibitor Cocktail (PIC, 1:200, Sigma), 2 mM EDTA). The column was washed extensively with wash buffer, and then elution was performed in wash buffer containing

10 mM maltose. For pry109, peak fractions were pooled, the MBP-tag was cleaved with TEV protease overnight without rotation and then loaded on a Superdex 75 Increase 10/300 GL column pre-equilibrated with SEC buffer (25 mM HEPES, 300 mM NaCl, 10% glycerol, 2 mM EDTA). For pry99, peak fractions were pooled and loaded on a Superdex 200 Increase 10/300 GL column pre-equilibrated with SEC buffer (25 mM HEPES, 300 mM NaCl, 10% glycerol). For pry94, peak fractions were dialyzed overnight in dialysis buffer containing 25 mM HEPES, 300 mM NaCl, 10% glycerol. Fractions containing protein were concentrated in 10 kDa MWCO Amicon ultra centrifugal filters (Millipore), aliquoted, snap froze in liquid nitrogen, and stored at −80 °C.

Expression and purification of [eGFP]Mer2, and [Alexa594]Mer2 were performed as previously described[9].

For expression of recombinant Smt3[6xHis] in E. coli, pCCB998 was transformed in BL21 (DE3)pLysS cells and plated on LB plates containing kanamycin. Cells were then cultured in LB media at 37 °C to an OD$_{600}$ of 0.6. Expression was carried out for 3 hours at 37 °C with 1 mM IPTG. Cells were centrifuged at 18 °C at 4000 g for 15 mins and were directly resuspended in lysis buffer (20 mM NaPi, pH 6.5, 30 mM imidazole, 350 mM NaCl, 0.1 mM DTT, 1 mM PMSF). Cells were lysed by sonication (10 W, 4 mins, 4 sec pulse) and centrifuged at 38,000 × g for 20 mins. Soluble fraction was incubated for 1 h with 1.5 ml Ni-NTA resin, pre-equilibrated with wash buffer (20 mM NaPi, pH 6.5, 30 mM imidazole, 350 mM NaCl, 0.1 mM DTT, 0.1 mM PMSF). The column was washed extensively with wash buffer and then eluted in wash buffer containing 500 mM imidazole.

For the production of the doubly labeled U-[$^{13}$C, $^{15}$N] Smt3[6xHis] protein, the IPTG induction was carried out in minimal medium containing M9 salts (6.8 g/L Na$_2$HPO$_4$, 3 g/L KH$_2$PO$_4$, and 1 g/L NaCl), 2 mM MgSO$_4$, 0.2 mM CaCl$_2$, trace elements (60 mg/L FeSO$_4$·7H$_2$O, 12 mg/L MnCl$_2$·4H$_2$O, 8 mg/L CoCl$_2$·6H$_2$O, 7 mg/L ZnSO$_4$·7H$_2$O, 3 mg/L CuCl$_2$·2H$_2$O, 0.2 mg/L H$_3$BO$_3$, and 50 mg/L EDTA), BME vitamin mix (Sigma), and 1 g/L $^{15}$NH$_4$Cl and 2 g/L [$^{13}$C$_6$]glucose (CortecNet) as the sole nitrogen and carbon sources, respectively. The purification protocol remained the same as described above.

## In vitro condensation assays

Proteins were diluted to a 10 × stock of their appropriate working concentrations in their respective storage buffers. Reactions were performed in a buffer containing 25 mM HEPES-HCl (pH 7.5), 2 mM DTT, 1 mg/ml BSA, 5% glycerol, 5 mM MgCl$_2$, 5% PEG-3350, and NaCl. Considering the salt contributed by the protein dilution buffer, the final concentration of NaCl in a reaction was adjusted to 120 mM. Unless specified otherwise, all reactions contained 400 nM [Alexa488]Mre11 or 100 nM [eGFP]MRX, 5.7 nM plasmid substrate, 5 % PEG-3350, and were imaged 30 minutes after assembly. A typical 20 μL binding reaction contained 2 μL protein of 10 × stock of indicated concentration, 10 μL of 2 × reaction buffer, and 150 ng of supercoiled pUC19 (5.7 nM). Typical reactions were assembled at 30 °C for 30 mins with gentle mixing every 5 minutes, unless mentioned otherwise. 5 μL was dropped onto a microscope slide and covered with a coverslip. All images were captured on a Zeiss Axio Observer with a 100 ×/1.4 NA oil immersion objective except for images provided in Fig. 4G, which were captured on Leica Stellaris DMI 8 confocal microscope with a 63x/1.2 NA water immersion objective. Images were analyzed with ImageJ using a custom-made script[9]. In brief, 129.24 × 129.24 μm (2048 × 2048-pixel) images were thresholded to mean intensity of the background plus three times the standard deviation of the background. Masked foci were counted, and the intensity inside the focus mask was integrated. Data points represent averages of at least 8-10 images per sample. Data were analyzed using Graphpad Prism 10.4.0.

## Gel shift assays

Proteins were diluted to their appropriate working concentrations in their respective storage buffers. A typical 20 μL binding reaction was

performed in a reaction buffer containing 25 mM HEPES-HCl (pH 7.5), 2 mM DTT, 1 mg/ml BSA, 10% glycerol, 5 mM EDTA, and NaCl adjusted to a final concentration of 100 mM, 1 nM pUC19 plasmid substrate, and the indicated concentration of protein. Reactions were assembled at 30 °C for 30 mins and resolved in a 1% agarose (SeaKem LE Agarose, Lonza) at 60 V for 120 mins at 4 °C. Gels were stained with SYBR Gold Nucleic Acid Gel Stain (S11497, Invitrogen) for 40 mins and visualized with Amersham Typhon biomolecular imager (Cytiva).

## Yeast targeting vectors and strain construction

Yeast strains are generated from *Saccharomyces cerevisiae* SK1 background and are listed in Supplementary Table 4.

To produce a yeast targeting vector, *MRE11-myc8::URA3* was amplified using cb1424 and cb1425 from the genomic DNA of CBY375 (SKY1361) and cloned into the TOPO vector by TOPO blunt cloning, generating pry2. Plasmids to produce *mre11-ΔIDR* (pry30), *mre11-ΔC49* (pry24), and *mre11-ΔC15* (pry42) mutants were generated by inverse PCR followed by self-ligation of pry2 using primers pp24 and pp25, cb1435 and pp46, and pp72 and pp73, respectively. Plasmids for the *mre11-SIM* (pry56) and *mre11-SIM + ΔC15* (pry57) were generated similarly using primers pp3 and pp4 and templates pry2 and pry42, respectively. Genomic integration of wild-type and truncated *MRE11-myc8::URA3* cassettes was performed by SpeI and NotI digestion of the corresponding plasmids and insertion in the endogenous *MRE11* locus of strain CBY006 by 'LiAc'-based transformation.

Plasmids to produce *MER2* mutant strains were based on pMH002, which contains a *MER2::HphMX* cassette cloned into a TOPO-based vector, as described[45]. To generate *mer2-EQEK*, pMH002 was amplified by inverse PCR followed by ligation using primers cb1561 and cb1562 to yield pCCB1046. An internal V5 (iV5) tag was introduced between Mer2 residues 248 and 249 (Mer2[iV5]) using primers dam005 and dam006 on plasmid backbone pMH002 by inverse PCR followed by ligation to generate pDAM003. *mer2[iV5]-EQEK* was constructed by inverse PCR followed by ligation of pDAM003 using primers cb1451 and cb1562. Genomic integration of *mer2-EQEK::HphMX* and *mer2[iV5]-EQEK::HphMX* cassettes was performed by SpeI, NotI, and XmaI digestion of the respective plasmids and insertion into the endogenous *MER2* locus of CBY006 by 'LiAc'-based transformation. The *MER2[iV5]::HphMX* allele was constructed similarly, following BamHI and SphI digestion of pDAM003.

All strains were genotyped by PCR and sequencing, and the opposite mating type was generated by crossing with CBY007. All other yeast strains were generated by crossing with appropriate genotypes listed in Supplementary Table 4.

## Spore viability assay

For the spore viability assay, a small patch of the diploid strain was incubated in sporulation media (2% potassium acetate) at 30 °C, 250 rpm for two days. After two days, 1 mL of sporulating culture was centrifuged, and all but 200 μL of supernatant was removed. To digest yeast cells, 2 μL of concentrated sporulation culture was mixed with 100 μL 1 M sorbitol and 1 μL 10 mg/ml zymolyase and incubated at 30 °C for 21 mins. 20 μL of digested cells were dropped on a YPD plate, left to dry for about 10 mins, and were micromanipulated using a tetrad dissector (MSM400, Singer Instruments). At least 20 tetrads were dissected for each assay, and spore viability was assessed by calculating the number of viable spores after 2 days of incubation at 30 °C.

## Yeast culture and meiotic synchronization

Following standard protocols, strains were patched on YPG plates, mated and streaked on YPD plates, and selected diploid colonies grown in liquid YPD at 30 °C, 250 rpm, overnight. For meiotic synchronization, diploid cultures grown overnight in YPD were transferred to YPA (1% yeast extract, 2% peptone, 1% potassium acetate) at OD$_{600}$ 0.2 and grown for 12–14 h at 30 °C, 250 rpm. Once the cultures

reached OD$_{600}$ between 1.2–1.6, cells were washed once with pre-warmed sterile water and immediately transferred to sporulation medium supplemented with amino acids (320 μL amino acid complementation media for 100 mL of sporulation media (SPM)) and were kept shaking at 30 °C, 250 rpm during the entire meiotic time-course. For MMS and CPT (Sigma) sensitivity assays, serial dilutions of overnight cultures were spotted on freshly prepared YPD-MMS or YPD-CPT plates containing the indicated percentage of MMS or CPT, respectively. Plates were grown for two days at 30 °C. For immunofluorescence of vegetatively growing cells, overnight cultures were refreshed by diluting to OD$_{600}$ 0.2 and grown for 3-4 hours to reach OD$_{600}$ 1.2–1.4 before subjecting to MMS or MMS followed by 5% 1,6-hexanediol treatment. For 5% 1,6-hexanediol treatment, cells were first converted to spheroplasts and then treated with 5% 1,6-hexanediol for 4-5 min. Spheroplasts were then immediately washed, lysed, and fixed using the protocol described below.

## Spreading and immunofluorescence of yeast nuclei spreads

Meiotic cultures were harvested 4 h after transfer to SPM, washed with sterile cold water, and resuspended in 1 M sorbitol, 1 × PBS (pH 7), 10 mM DTT, 0.5 mg/ml zymolyase 20 T, and incubated for 30 mins at 30 °C with gentle shaking. Spheroplasts were collected by centrifuging for 1 min at 1500 rpm and were washed gently with ice-cold 0.1 M MES-1 M sorbitol. Spheroplasts were then centrifuged, lysed by adding ice-cold 0.1 M MES and 4% paraformaldehyde, followed by vigorous finger-vortexing and immediately fixing on microscopy slides for 1 h at room temperature. Slides were washed three times with 1 ml 0.4% PhotoFlo 200 solution (Kodak), air dried and stored at − 20 °C or directly used for processing.

Slides were blocked with 90% FBS, 1 × PBS for 1 h at room temperature in a humid chamber and then incubated with primary antibody (mouse mAb anti-myc, 1:200 (2276S, Cell Signaling Technology), rabbit anti-phospho H2A-S129, 1:200 (ab15083, abcam)) diluted in 3% BSA, 1 × PBS in a humid chamber for 2 h at 37 °C or overnight at 4 °C. Slides were washed three times with 1 × PBS in a Coplin jar, were incubated with secondary antibody (goat anti-mouse IgG Alexa Fluor™ Plus 488, 1:200 (A32723, Invitrogen), donkey anti-rabbit IgG Alexa Fluor™ 546, 1:200 (A10040, Invitrogen)) diluted in 3% BSA, 1 × PBS in a humid chamber at 37 °C for 1 h. Slides were washed in the dark three times for 5 mins with 1 × PBS, mounted with Vectashield containing DAPI (Vector Labs). Images were captured on a Zeiss Axio Observer with a 100 ×/1.4 NA oil immersion objective and were analyzed in ImageJ.

## Co-immunoprecipitation

Meiotic cultures were collected at 4 h after transferring to SPM, washed once in ice-cold 1 × PBS containing 1 mM PMSF, following which, they were transferred to 1.5 ml screw-cap tubes and were either snap frozen in liquid nitrogen or processed directly. Cell pellets were resuspended in 600 μl of IP buffer composed of 25 mM Tris-HCl, pH 7.5, 150 mM NaCl, 1 % Triton × 100, 5 mM EDTA, 0.1% sodium deoxycholate, and protease inhibitors (1 mM PMSF, PIC Sigma 1:100, 1 μg/μl leupeptin, 125 U/ml benzonase, 1 μg/μl aprotinin). 900 μL of 0.5 mm zirconium beads were added to the cell pellet, and cells were lyzed in a Precellys bead beater at 7200 rpm for 12 × 15 seconds while chilling the tubes for 2 min on ice between each cycle. Cell extract was collected by poking a hole at the bottom of the tubes while placing the tubes on top of another 1.5 ml tubes and centrifuging at 800 × g for 1 min at 4 °C. Beads were washed with 500 μl of IP buffer and collected in the same tube. Cell lysate was spun again at 800 g for 5 min at 4 °C and supernatant was transferred to 15 mL tubes. Sonication was performed three times in ice in Bioruptor Plus at intensity (M) in a 30 s ON / 30 s OFF cycle for 15 min each time. The lysate was centrifuged at 15.000 g at 4 °C for 10 min and the supernatant (∼ 800 μl) was transferred to new 1.5 ml tubes. 80 μL (10%) was collected as Input fraction and snap

frozen. 1 µL of mouse monoclonal anti-myc primary antibody was added to each tube, and tubes were rotated overnight at 5 rpm at 4 °C. The following day, Dynabeads Protein G (30 µl per tube) were added to Protein LoBind tubes and washed two times with 1 mL IP buffer containing 1 mM PMSF using DynaMag2. The lysates with the antibody were transferred to the tubes containing Dynabeads and were rotated for 3-4 hours at 5 rpm at 4 °C, following which, the Dynabeads were collected and washed four times with 1 ml IP buffer containing 1 mM PMSF. Finally, 60 µL IP buffer and 20 µL 4 × Lammelli buffer was added to the tubes and incubated for 5 mins at 95 °C while shaking at 1000 rpm. TCA extraction was performed on the input fractions by adding 8 µL 100 % TCA (final 10 % TCA) and incubating on ice for 30 mins. Precipitated proteins were solubilized in 40 µL 1 × Laemmli sample buffer incubated for 5 mins at 95 °C. Immunoblotting was performed using the protocol described below.

### Western blotting of yeast meiotic extracts

Meiotic cultures at desired timepoints were harvested, washed in ice-cold water, and lysed in 20% trichloroacetic acid (TCA) by agitation in a bead beater (Precylles Evolution) using 0.5 mm zirconia/silica beads. Precipitated proteins were solubilized in Laemmli sample buffer, and appropriate amounts of protein were separated by SDS-PAGE and analyzed by Western blotting. Western blotting was performed using standard protocol. Primary antibody used was mouse monoclonal anti-myc at 1:1000 dilution (2276S, Cell Signaling Technology), mouse monoclonal anti-V5 at 1:500 (R96025, Invitrogen), mouse monoclonal anti-PGK1 at 1:5000 (ab113687, Abcam) and secondary antibody used was goat anti-mouse IgG-HRP conjugated at 1:10,000 dilution (AP308P, Chemicon). Western blots were revealed using SuperSignal West Femto Maximum Sensitivity Substrate (Thermo Scientific) in Amersham Imager 600 (Cytiva).

### Southern blotting

Meiotic DSB analysis by Southern blotting was performed as previously described[75]. In brief, synchronized cultures undergoing meiosis were collected at the indicated time points. After DNA isolation, 1 µg of genomic DNA was digested by PstI and separated on a 1% TBE-agarose gel. DNA was transferred to Amersham™ Hybond™-N + nylon membranes (Cytiva) by vacuum transfer, hybridized with GAT1 probe (amplified with primers: 5′-CGCGCTTCACATAATGCTT CTGG, 5′-TTCAGATTCAACCAATCCAGGCTC) and developed by autoradiography.

### Pull-down assay

Wild-type and mutant MBP-tagged Mre11-C49 and HisSUMO-tagged Mer2 were co-expressed in 50 mL of E. coli BL21 cultures and purified by affinity chromatography on Ni-NTA resin following a procedure similar to that described above. Briefly, cells were lysed by sonication and centrifuged at maximum speed for 30 mins at 4 °C on a table-top centrifuge. A small fraction of the supernatant was collected as 'Input' and the remainder was incubated with 120 µL Ni-NTA resin, pre-equilibrated with wash buffer (25 mM HEPES pH 7.5, 20 mM imidazole, 0.1 mM DTT, 0.1 mM PMSF, 10% glycerol), for 1 hour on a rotating wheel at 4 °C. The resin was washed twice with 2 ml in batch and five times with 2 ml on column before eluting with 250 µL of 500 mM imidazole in wash buffer. Input and elution fractions were separated by SDS-PAGE followed by immunoblotting with primary murine anti-MBP monoclonal antibody at 1:10,000 dilution (E8032S, NEB) and secondary goat anti-mouse IgG-HRP conjugated at 1:10,000 dilution (AP308P, Chemicon). Western blots were revealed using SuperSignal West Femto Maximum Sensitivity Substrate (Thermo Scientific) in Amersham Imager 600 (Cytiva).

### Nuclear magnetic resonance spectroscopy

The samples contained 0.3–0.6 mM of U-[$^{13}$C,$^{15}$N] Smt3 in 20 mM sodium phosphate, 20 mM NaCl (pH 6.5), 0.02% NaN$_3$ and 10% D$_2$O for the lock. All NMR spectra were acquired at 298 K on a Bruker Avance III HD 800 MHz spectrometer, equipped with a TCI cryoprobe. The NMR data were acquired in TopSpin 3.6 (Bruker) and processed in NMRPipe v.7.5[76] and analyzed in CCPNMR v. 2.4.2[77]. Assignments of Smt3 backbone amide resonances were taken from literature[78] and verified by 3D HNCACB, HN(CO)CACB, and $^{15}$N-edited NOESY-HSQC (mixing time 100 ms) spectra. Further assignments of methyl resonances were obtained from [$^1$H, $^{13}$C] HSQC, 3D HBHA(CO)NH, and (H)CCH TOCSY experiments performed on the wild-type SIM-bound Smt3 sample, which exhibited superior spectral quality compared to that of the free protein. The assigned $^1$H, $^{13}$C and $^{15}$N chemical shifts of the free and bound Smt3 have been deposited in the Biological Magnetic Resonance Bank (http://www.bmrb.wisc.edu/) under the accession number 53209.

The NMR binding experiments were performed by an incremental addition of wild-type or mutant SIM peptides (1.2 mM stocks in the working buffer) to 0.3 mM samples of U-[$^{13}$C,$^{15}$N] Smt3, with the spectral changes monitored in [$^1$H,$^{15}$N] HSQC spectra acquired at each increment and [$^1$H, $^{13}$C] HSQC at the end of titration. The average chemical shift perturbations ($\Delta\delta_{avg}$) were calculated as $\Delta\delta_{avg} = (\Delta\delta_X^2/n + \Delta\delta_H^2/2)^{0.5}$, where $\Delta\delta_X$ and $\Delta\delta_H$ are the chemical shift changes of the backbone amide nitrogens or methyl carbons ($\Delta\delta_X$) and protons ($\Delta\delta_H$) of Smt3 residues upon addition of 1.2 molar equivalents of SIM peptides, and $n = 50$ or 9 for the backbone amide and methyl groups, respectively.

### Peptide synthesis

The peptides were synthesized by standard Fmoc-based solid-phase peptide synthesis. Preloaded Fmoc-Glu-Wang resin (0.55 mmol/g) was swollen in N,N-dimethylformamide (DMF) and the Fmoc-deprotection steps consisted of shaking the resin in two consecutive steps of 5 min and 15 min, in 20% 4-methylpiperidine in DMF containing 0.1 M 1-hydroxybenzotriazole (HOBt). The coupling steps were performed with conventional Fmoc-protected amino acids (3 equiv.) (except for N-α-Fmoc-(O-3-methyl-pent-3-yl)aspartic acid), hexafluorophosphate benzotriazole tetramethyl uronium (HBTU, 3 equiv.) and N,N-diisopropylethylamine (6 equiv.). After synthesis of the full sequence on resin, the peptide was cleaved with trifluoroacetic acid (TFA)/triisopropylsilane (TIS)/H$_2$O (90/5/5, v/v/v). The products were purified by preparative reverse-phase HPLC using an acetonitrile/water eluent mixture containing 0.1% TFA.

### Isothermal titration calorimetry

The measurements were performed in 20 mM NaPi 100 mM NaCl pH 7.5 on a Microcal ITC200 calorimeter at 25 °C. The syringe contained 2 mM of peptide, while the cell held 150 µM or 300 µM of protein. Given the poor peptide solubility in aqueous buffers, the peptide solutions necessitated the addition of 10% DMSO. For the ITC titrations, equal amounts of DMSO (10%) were included in both compartments to minimize the heat of dilution. Each titration consisted of a first injection of 0.4 µL, followed by 12-13 injections of 2 µL peptide into the cell, separated by intervals of 120 s. The first injection was discarded during the analysis of the data. The wild-type peptide titration on Smt3 was performed in duplicate. The Microcal LLC ITC200 Origin software was used to fit the data to a single binding site model.

### Yeast two-hybrid

Bait and prey plasmids were co-transformed into yWL365 using the standard 'LiAc'-based transformation and plated on SC-Leu-Ura selective media. At least four independent transformants were tested for Mer2-Mre11 interaction by spotting a dilution series on SC-Leu- Ura-His + 25 mM 3-AT (3-amino-1,2,4-triazole) and growing for 4-5 days at 30 °C.

### Statistical analysis and data visualization

All statistical analysis and graphing were performed using Graphpad Prism version 10.4.0. Student's t test was used for the determination of

statistical significance and *P*-value calculation ($p \geq 0.05$, ns, not significant; **$p < 0.01$; ***$p < 0.001$; ****$p < 0.0001$).

## Reporting summary
Further information on research design is available in the Nature Portfolio Reporting Summary linked to this article.

## Data availability
The NMR data generated in this study have been deposited in the Biological Magnetic Resonance Bank with the accession number 53209. Accession codes for protein structures used for AlphaFold models are listed below: AF-P32829-F1, CAA60944, BAA02017, XP_003669210.1 [https://www.ncbi.nlm.nih.gov/protein/XP_003669210.1], XP_003672532.1 [https://www.ncbi.nlm.nih.gov/protein/XP_003672532.1], XP_001647040.1 [https://www.ncbi.nlm.nih.gov/protein/XP_001647040.1], XP_001642997.1 [https://www.ncbi.nlm.nih.gov/protein/XP_001642997.1], XP_003683996.1 [https://www.ncbi.nlm.nih.gov/protein/XP_003683996.1], XP_003686402.1 [https://www.ncbi.nlm.nih.gov/protein/XP_003686402.1] and AF-Q12306-F1. AlphaFold models are available in ModelArchive (modelarchive.org) with the following accession codes: ma-bjxr0 [https://modelarchive.org/doi/10.5452/ma-bjxr0], ma-sljg3 [https://modelarchive.org/doi/10.5452/ma-sljg3], ma-byd98 [https://modelarchive.org/doi/10.5452/ma-byd98], ma-itjlc [https://modelarchive.org/doi/10.5452/ma-itjlc] and ma-b9r3g. Source data are provided in this paper.

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

## Acknowledgements
We thank David Alvarez Melo for generating Mer2-iV5 tagged strains, John Weir for plasmids and strains, and C.C.B. laboratory members for discussion. We thank the Biological Imaging facility (IMABIOL) at UCLouvain and Marie-Christine Eloy for providing training in the use of the epifluorescence microscope. This work was supported by the European Research Council under the European Union's Horizon 2020 research and innovation program (ERC grant agreement 802525 to C.C.B.), and the Fonds National de la Recherche Scientifique (PDR grant T.0031.22 to C.C.B.). P.P. is funded by FNRS Aspirant fellowships (project 1.A908.22). C.C.B. is an FNRS Research Associate. W.E.Y.M. and S.B. acknowledge the Research Council of VUB for support through the Strategic Research Program SRP95 and the infrastructure grant OZR3939. NIH NIGMS grant R01GM074223 supported NH, who is also an Investigator of the Howard Hughes Medical Institute.

## Author contributions
P.P. designed, executed, and analyzed all experiments except as noted; M.S. performed Southern blot experiments; W.E.Y.M. synthesized peptides under the supervision of S.B., performed ITC experiments and analyzed NMR data; A.N.V. acquired and analyzed NMR data; R.B. performed yeast 2-hybrid experiments under the supervision of N.H.; C.C.B. supervised the research and secured funding. P.P. and C.C.B wrote the paper with input from all authors.

## Competing interests
The authors declare no competing interests.
