## [Transparent Peer Review file · Nature Communications]

Recruitment of Mre11 to recombination sites during meiosis

Corresponding Author: Dr Corentin Claeys Bouuaert

Version 0:

Reviewer comments:

Reviewer #1

(Remarks to the Author)

This paper aimed to characterize the mechanism(s) by which the Mre11 complex (MRX in budding yeast) promotes meiotic DSB formation in budding yeast. It has long been known that Mre11 is required for meiotic DSB formation by Spo11, a function that does not involve Mre11 nuclease activity and that involves the last 49 aminoacids of Mre11. Using a combination of in vitro biochemical and biophysical assays with recombinant proteins, AlphaFold modelling and yeast genetics, they bring novel insight into the functions of the Mre11 complex involved. The main conclusions are that Mre11, as well as MRX, can form DNA-dependent condensates in vitro, like the authors previously showed for the other DSB proteins, Mer2 and Rec114-Mei4. Mer2 condensates promote the nucleation of Mre11 condensates. They better characterize the previously described interaction between Mre11 and Mer2, using AlphaFold and mutation of the predicted interacting residues. The interaction is important for meiotic DSB formation and spore viability, although it is the only function explaining Mre11 requirement for DSB formation. Finally, they uncover a novel SUMO-interaction motif in Mre11, which, in addition to the last alpha-helix of Mre11 is required for DSB formation. These results are complementary to recent findings that showed that Mre11 also directly interacts with Spo11, to promote DSB formation. Altogether, although some results are still preliminary and the mechanism by which the MRX activates Spo11 is still not fully understood, this study brings novel important insight into the mechanisms by which the MRX promotes meiotic DSB formation, in particular through its interaction with Mer2. This study also reveals that meiotic DSB formation seems to be promoted, in vivo, by the formation of successive condensates, that may collaborate to concentrate Spo11 subunits and trigger its catalytic activity. Although the Mre11 complex is not required for meiotic breaks in mammals or plants, the general concept of condensates leading to meiotic DSB formation is of a broad interest.

Major comments:

- It would be important to examine Mer2 foci in vivo for the mer2-EQEK mutant, to measure if it is still forming normal foci. Especially important since its phenotype seems to be different from that of mre11 Δ C15.
- In the same line, it would also be important to assess co-IP in meiotic cells between Mer2 and Mre11 Δ C15 and between Mer2-EQEK and Mre11.
- Figure S6A: the authors should repeat this experiment and add levels of a control protein, such as Pkg1, in order to say that mre11 Δ C15 levels were not changed. They do seem reduced here.

-

Other comments:

- In Figure S2A: why is the band corresponding to Xrs2 more intense than the two others? This would need some explanation.
- Figure S1C: the x axis seems to be wrong. It is rather time (minutes).
- Page 4 line 104: should be "Mre11 and MRX condensation...crowding agents (Figures 1E, 1F and S2B)"
- Figure 1B and 1D: it appears clearly that for the MRX, the total foci intensity is nearly zero after DNase treatment, whereas for Mre11, the total intensity is more or less preserved. So it does not seem that the remaining foci tend to accumulate more protein in the case of MRX. How to explain these results?
- Figure 1J: focus number seems to be zero without treatment, because of the y axis scale. Can the authors give the actual numbers (in the text, or in the Figure legend)?
- Page 5 line 124: "...important for meiotic DSB formation". It could be added that this region is also required for Mre11 binding to DSB hotspot (ref 40)
- Page 5 line 137 and Figure S3C, D and F: clarify what "Supershift" means ("complexes of reduced mobility"?)
- Page 5 line 151 "...dissolved MMS-induced foci" and Figures 3A and B: the authors could remind that this is after 4-5

minutes 1,6-hexanediol treatment (as indicated in the methods section)

- Page 6 line 175 and Figure 4E: the authors should show the data also for mer11 Δ C49

- Figure S4F: in the absence of quantification, this experiment should be remade since there is almost no Mre11-Myc signal in the 0h time-point.

- Page 7 and Figure 4F: for the Δ IDR and the Δ C49 it seems difficult to assess the colocalization since Mre11 foci are almost absent.

- Page 6 line 208: "implicated in Mre11 binding": should refer to Figure S5B.

- Page 6 line 225: typo: should be "...Mer2 by approximately 45%"

- Page 9 line 271: "delayed and reduced DSB formation": from the two experiments shown in Figure 7B and Figure S10C, DSBs are delayed, but not reduced. This should be corrected.

- Figure S10C: the right title above the Southern blot should be mre11-SIM3 and not mre11- Δ SIM3

Reviewer #2

(Remarks to the Author)

This article builds on a previous study published in Nature in 2021, in which the authors reported that the yeast RMM protein complex assembles into nucleoprotein condensates along meiotic chromosomes, where it recruits Spo11 to stimulate DNA cleavage. In this new article, the authors characterize the role of the MRX complex in meiotic DSB formation. It was already known that the Mre11 C-terminus is required for meiotic DSB formation, and directly binds to several regions of Spo11 (as shown by pulldown and MST). Here, they show that in their in vitro conditions, Mre11 as well as the MRX complex are able to form condensates in the presence of DNA, and they identify that condensation relies on DNA binding but also on the presence of the Mre11 C-terminus that is not strongly involved in DNA binding but probably in uncharacterized protein-protein interactions. In cells, formation of meiotic foci of Mre11 depends on other DSB proteins. The authors having already shown that the RMM protein complex assembles into condensates, they tested if these condensates can recruit Mre11. They found that indeed, in cells with defective RMM condensation, the formation of Mre11 foci is strongly reduced. Moreover, they detect by yeast 2-hybrid a direct interaction between the RMM protein Mer2 and Mre11, and propose that this interaction is responsible for recruiting Mre11 in RMM condensates. Similarly, when expressing a Mre11 mutant deleted from all its C-terminal IDR region, the formation of Mre11 foci is strongly reduced. Deletion of the Mre11 C-terminus known to favor MRX condensation only decreases the number of MRX foci but not their intensities in cells. It also decreases colocalization with Mer2. The authors propose that the direct interaction between Mer2 and Mre11 mediates the nucleation of MRX condensates, whereas the propensity of MRX to form condensates is responsible for the growth of the condensates. Finally, the authors try to understand how Mre11 binds to Mer2, based on AlphaFold calculations, pull-down assays and immunofluorescence experiments. They study how much sumoylation regulates this interaction, using NMR, ITC, and immunofluorescence experiments.

Figure 1 clearly shows that, in the tested conditions, Mre11 and MRX form condensates and that DNA favors nucleation of these condensates. In order to understand the molecular mechanisms of condensation, the authors tested the role of the different Mre11 regions in Figure 2. They first identified the different regions of Mre11. Therefore, in panel A, they present an AlphaFold Model of yeast Mre11. As the experimental 3D structure of this protein is available (PDB ref: 9BI4), it would be more reliable to present this structure. Moreover, in panel B, the authors present a prediction of the disorder as a function of the Mre11 sequence. Here again, from the analysis of the experimental structure, it is possible to identify the disordered regions of Mre11 bound to Rad50 and DNA: the cryo-EM structure of Mre11 is well-defined between residues 1 and 412 (phosphodiesterase and capping domains), and then between residues 442 and 522 (Rad50 binding domain). Such an analysis provides a precise definition (more reliable than a prediction) of the IDRs of yeast Mre11.

In Figure 2 panels C, D and E, the authors show that deleting most of the phosphodiesterase domain (Del 10-270) or the capping domain, the linker between the capping and Rad50 binding domains, and part of the Rad50 binding domain (Del 290-472) decreases the number of foci, whereas deleting the C-ter IDR or the C-ter 49 last residues decreases the intensity of foci (but strongly increases their number) in vitro. So, the folded domains contribute to foci nucleation, whereas the C-ter IDR contributes to the growth of the condensates. When the same experiment is performed with the whole MRX complex, deletion of the C-ter 49 last residues decreases both the intensity and the number of foci, so that it seems that the C49 region is involved in both the nucleation and the growth of the condensates. Can the authors comment on that?

More generally, I wonder how it could be checked that the truncated variants of Mre11 are soluble, especially in the absence of Rad50. In Figure S3B, the authors show a convincing CBB picture of these different variants that have been successfully purified. However, truncating part of the folded domains, as in Del 10-270 and Del 290-472, creates exposed hydrophobic residues that might favor Mre11 aggregation, thus modifying the condensation mechanisms.

Also, which interactions between Mre11 monomers could contribute to foci nucleation and growth? Does Mre11 dimerization contribute to the multivalent interactions involved in condensation ?

At this point, as the presence of DNA and the folded domains contribute to the nucleation of condensates, the authors characterize the DNA binding properties of their different Mre11 variants using EMSA. Here, as a non Mre11 expert, I don't understand how Mre11 binds to DNA. Indeed, in the cryoEM structure of Mre11/Rad50/dsDNA (PDB ref: 9BI4), Mre11 does not contact DNA. Moreover, yeast Mre11 is an acidic protein (IP around 5), only its C-terminal IDR is positively charged (the IP of its 49 Cter residues is particularly high: around 12). In the cryoEM structure of Mre11/Rad50/dsDNA, there are a few small positively charged patches at the surface of Mre11, however they are at the interface with Rad50. Was it previously reported that yeast Mre11 interacts with DNA? Does this happen in the activated state of Mre11? But then, is the interaction

with Rad50 important for Mre11 to bind to DNA? Can the authors comment this point?

In Figure 4, the authors observed the Mre11 foci in meiosis. They found that the 49 Cter residues are important for foci nucleation, which was not observed in vitro. Thus, additional cellular partners account for this nucleation effect. They propose that the RMM complex is able to trigger nucleation of Mre11 foci. To test this in vitro, they produced condensates of Mer2 and Mre11, and found, by mixing these condensates, that the two proteins co-localize. This colocalization is mildly favored in the presence of the folded domains, but strongly depends on the presence of the IDR region of Mre11, and in particular its 49 Cter residues. How do the authors explain that the Mre11-C49 construct accumulates around Mer2 foci, but does not enter into these foci?

In Figure 5, the authors attempt to obtain a structural description of the interface between Mre11 and Mer2. They provide an AlphaFold2 model of the complex between 4 Mer2 molecules and 4 Mre11 C49 peptides. Indeed, they showed in a previous paper that Mer2 is a coiled-coil protein forming a tetramer (based on SEC-MALS and SAXS data). I found no Material and Methods section related to the AlphaFold2 calculations, and no scoring of the models (pLDDT plots; TM and ipTM scores). In Figure S5A, the heat maps suggest that the predictions are unreliable. Moreover, AlphaFold2 is known to poorly predict coiled-coil containing structures. From my point of view, if the AlphaFold2 models are poorly scored, they should not be shown. The authors can make hypotheses on the interfaces independently of the AlphaFold2 results. But showing / discussing AlphaFold2 models that are provided with low confidence scores is misleading: the reader is tempted to believe in these models, forgetting that they are based on no data. I understand that the authors found, by analyzing their models, that the motif SSM1 of Mer2 is involved in Mre11 binding, and this was previously shown elsewhere. But models obtained with poor scores do not reinforce the experimental data. As the authors can purify Mer2 and buy a synthetic peptide of Mre11 corresponding to C49, why didn't they use SEC-MALS and SAXS to characterize the Mer2-Mre11 complex, as previously reported for Mer2 alone? Is it because "the interaction is relatively weak" ? What do "relatively weak" mean ? Could the authors use ITC or BLI to measure the affinity between Mer2 and Mre11 ?

In order to identify the residues at the interface between Mer2 and Mre11, the authors present pulldown assays revealed by WB. Could it be added to the legend of Figure5B that the input and pulldown are revealed by WB using Ab against MBP? It is in the Material & Methods section, but it is essential for understanding the figure panel. There are a lot of not reproducible interaction data obtained by pulldown revealed by WB in the literature. To keep this data as a main figure panel, I would recommend to confirm this experiment using another experimental method. As for ex by BLI, coating a synthetic biotinylated Mre11-C49 peptide on streptavidin beads and adding increasing concentrations of HisSUMO-Mer2. Or by MST using a fluorescent Mre11-C49 peptide.

As sumoylation sites were mapped to the SSM1 region of Mer2, potentially involved in Mre11 binding, the authors searched for the impact of sumoylation on Mer2-Mre11 binding in cells. They detected 3 SUMO-interacting motifs in Mre11, two in the phosphodiesterase domain and one in the Cter IDR, just before the last 49 residues (633-637). They used AlphaFold2 to predict that yeast Smt3 binds to the third SIM (SIM3). Here again, no scoring is provided to judge about the robustness of the prediction. However, a careful NMR analysis (though at low salt concentration) demonstrated that the SIM3 peptide specifically binds to the SIM-binding groove of Smt3. The ITC analyses provided the Kd of the interaction. In Figure S9, the information on concentration is misleading: only one concentration is written so that it is unclear if this is the concentration of the Smt3 protein or the peptide. Could the authors add the control obtained by injecting the WT peptide into the buffer?

In general, as there is no proof that in cells SIM3 SUMOylation is important for meiosis, and as the impact of sumoylation of SIM3 on the interaction between Mer2 and Mre11 is still unclear, the analysis of the impact of sumoylation does not strengthen this study and could be moved to the supplementary figures.

Discussion:

In general, the discussion is based on experimental data that are still to be strengthened. When discussing about "the Mer2 alleles previously reported to reduce Mre11 interaction that involve amino acids predicted to point inside the Mer2 tetrameric coiled coil", why should Mer2 stay in its tetrameric state upon binding to Mre11 ?

Minor points:

1. Condensation: in the Mat & Met, the conditions used to study condensation include PEG (these conditions are precisely: 25 mM HEPES-HCl (pH 7.5), 2 mM DTT, 1 mg/ml BSA, 5% glycerol, 5 mM MgCl₂, 5% PEG-3350, and 120 mM NaCl), whereas in the legend of Figure 1E,F, the impact of adding PEG is shown. Is there PEG in the dilution buffer only for the experiments displayed in Figure 1E,F ?
2. NMR: in the last sentence of the Mat & Met sections, there is an n value that is cited but not present in the equation used for the CSP analysis.
3. NMR and ITC: why is the salt concentration so different between the NMR and ITC conditions ? The ITC conditions are closer to physiological conditions than the NMR conditions. Is the salt concentration influencing the interaction ?

Reviewer #3

(Remarks to the Author)
NCOMMS-25-61398-T

Recruitment of Mre11 to recombination sites during meiosis

In this manuscript, Priyadarshini et al. reported that a widely studied DSB end processing factor, Mre11 of the MRX complex, forms condensate at DSB foci in both vegetative cells and during meiosis of *S. cerevisiae*, and this involves a C-terminal IDR and the last C-terminal 15 amino acids that bind Mer2 and SUMO modification. Interestingly, while the condensate is dispensable for vegetative DSB repair, it is essential for meiosis. While some results are interesting and may offer new insights into the role of Mre11 in *S. cerevisiae* meiosis, the main conclusion lacks strong support and significant issues remain unanswered.

Major issues:

1. The biggest issue is the unclearness of the true biological significance of the Mre11 condensate in DSB repair.
 - Why the condensate demonstrated different essentialities to vegetative DSB repair and meiosis? What is unclear is that if the condensates are part of the DSB foci and its formation involved the same protein-protein interaction principles (or same proteins and domains are involved), why are they essential for meiosis but not for DSB repair? What determines the different essentialities here?
 - This also raises additional concerns as to the importance of the Mre11 condensate concept in vegetative DSB repair. If it is not important, then what does it matter, or worse, who cares this condensate formation?
 - Then how to separate this condensate formation from the normal Mre11 foci at DSB repair site in vegetate cells?
 - While a DSB repair assay was performed for meiotic cells, it was absent for vegetate cells. Only a survival result was shown for vegetate cells expressing Mre11 mutants. Should also examine if these condensate deficient Mre11 mutants impact DSB repair in vegetate cells.
 - If condensate is essential for meiosis, can you rescue the lethality issue of the strains with Mre11-IDR or C49 deleted? Are there any analyses to see if the Mre11-IDR or the C49 residues alone can form foci in meiotic cells?

Without clear answers to these questions, the true biological significance of the Mre11 condensates at DSB sites remain uncertain.

2. How exactly Mre11 condensates are formed, regulated, and contributed to DSB foci in vegetate and meiotic cells is unclear.

- First, based on data presented, there are five regions/motifs of Mre11 shown to be involved: the IDR (~520-677), the C49 (residues 643-692), the C15 (residues 677-692), and the SUMO binding region (residues 633-637). The C15 binds to Mer2 and deletion of which caused similar level of foci reduction as the C49 deletion; yet, C15 is not essential for meiosis essentiality, whereas the C49 does. These results suggest the last 34 residues (643-677) of the IDR are important for meiosis. However, a comprehensive comparison of these above five fragments on in vitro condensate assembly with or without Mer2 is lacking. Also, how these fragments contribute to Mre11 foci formation in vegetate vs meiotic cells are unclear.
- Second, there is the lack of DSB repair in vegetate vs meiotic cells expressing Mre11 mutants with deletion of just the above five fragments. Similarly, meiotic cell viability of these mutants is important to show.

3. While the manuscript presented some interesting findings, often there was no discussion regarding the biological/pathological implications/significance of the findings.

- Fig 3B has no discussion regarding the increase in the percentage of Mre11 aggregates by 1,6-HD treatment and the relationship of this increase to the reduction in Mre11 foci.
- Fig 4A showed no Mre11 aggregates by 1,6-HD, which is different from vegetate cells and somehow may indicate a difference between these two states. But then, it did so when expressed in meiotic mer2-KRRR cells in Fig 4D. So, going back to the point above, what are exactly the difference between these foci and the large-in-size aggregates of Mre11? Are these reflecting real biological/pathological changes or just expression-induced nuance of Mre11 under different background? This also ties to the major issue #1.

Minor issues:

- Fig 1B & 1D: it stated that "While nuclease treatment strongly reduced the numbers of Mre11 and MRX foci, the minority of Mre11 foci that resisted nuclease treatment tended to accumulate more protein". Where is the evidence that the residual condensate accumulated more protein? The selected representative images clearly did not support that. The quantitative intensity data in Fig 1B and 1D showed much reduced fluorescence intensities, which reflect the abundance of proteins in any condensate, for all foci (including the nuclease resisted ones). None of the quantitated foci after nuclease treatment retained an intensity level even close to the lowest data point prior to nuclease treatment in these two panels. This statement clearly does not hold.
- Fig 4F was assembled with equal molar ratio of Mre11 and Mer2, then why a 4-fold ratio was used for Fig 4G (Mre11-C49:Mer2=4:1)? Will Mre11-C49 still coat Mer2 condensates if mixed at 1:1 ratio? In fact, if Mer2 works as a tetramer to recruit the MRX complex, it would be Mre11-C49:Mer2=1:4 in these in vitro assembly reactions.
- Fig 5A only modeled the C49 of Mre11. What will happen if a larger domain of Mre11 (like 620-692 used in Fig S7A) is used? This is important as the authors proposed multiple motifs of Mre11 binding to Mer2. There are a lot of back-and-forth information regarding the domains that are involved in Mer2 binding and condensate formation of Mre11, which were not clearly and systematically evaluated.
- Fig 7: The importance of the SIM3 motif in Mer2 binding in vivo is lacking. Why not performing similar pulldown assay as in Fig 5B for Mre11-SIM3 mutant and Mer2? Without this information, the model in Fig 7D and the conclusion in the text are premature.
- Some of the selected representative images are not impressive to support the conclusions. For instance, Fig 1A and S1C hardly tell any foci intensity increases. Fig 2C did not show any reduction in foci number for Mre11 Δ 290-472. The IDR deletion Mre11 mutant had much weaker foci than the C49 deletion in Fig 4B; yet, the quantitation showed the opposite.

- Figure legends are generally too simplified with much information missing. For instance, what was the conc of the eGFP-Mre11 protein in Fig S1B? What are conc of DNA in Fig 1A and 1C? What time point was the image taken for Fig 1A? Do Fig 2C-E contain a crowding agent? It would help the readers if all figure legends provided detailed information.
- "...focus formation of Mer2-EQEK was diminished compared to wild-type Mre11 by approximately 45% (Figure 5D)": Fig 5 D has nothing to do with Mer2 foci, but instead, it showed Mre11 foci under different backgrounds including in the Mer2-EQEK strain.

Reviewer #4

(Remarks to the Author)

Version 1:

Reviewer comments:

Reviewer #1

(Remarks to the Author)

In this revision, the authors have satisfactorily answered all my major and minor points.

Reviewer #2

(Remarks to the Author)

The authors answered to all my questions, I am fine with their answers, thanks.

Except for the addition of the BLI experiments (Suppl. Fig. 6e).

For me, it is mandatory to delete this panel. First, I don't see how you can measure an affinity of 1 mM by titrating a ligand between 12.5 and 25 μ M. Second, several essential controls are lacking, as titrating the analyte onto empty biosensors (Nickel-biosensors interact non-specifically with many ligands) and onto His-SUMO coated biosensors. Third, the legend is unclear, but, if, as I understood, the only displayed control experiments gave the less bright curves (titration onto the His-SUMO-Mer2 coated chip of MBP alone), then there is a huge non specific signal that should be subtracted from the main BLI curve.

For all these reasons, I think that this panel has to be removed.

Reviewer #3

(Remarks to the Author)

While the authors have addressed many technical points, my central concern remains largely unaddressed: the manuscript still fails to establish the biological significance of Mre11 condensates. Below, I focus specifically on this unresolved issue and the authors' responses to it.

Original comment: The biggest issue is the unclearness of the true biological significance of the Mre11 condensate in DSB repair.

- Why the condensate demonstrated different essentialities to vegetative DSB repair and meiosis? What is unclear is that if the condensates are part of the DSB foci and its formation involved the same protein-protein interaction principles (or same proteins and domains are involved), why are they essential for meiosis but not for DSB repair? What determines the different essentialities here?

Authors' response: Although the domains required for Mre11 foci formation in meiotic and vegetative conditions are the same, the assembly of these foci and their composition are fundamentally different. Vegetative Mre11 foci depend on DSBs, meiotic ones do not. Meiotic Mre11 foci depend on Mer2 and involve a host of other proteins that are absent from vegetative foci.

Note that we do not claim that Mre11 condensation is essential in meiosis. We claim that it cannot be essential in vegetative DSB repair (because deleting the IDR compromise condensation in vitro and foci formation in vivo with no detectable effect on MMS sensitivity).

Revision Comment: If the authors do not claim that Mre11 condensation is essential in meiosis, then what is the intended biological conclusion regarding these condensates? Under the authors' own framing, Mre11 condensation is neither essential for vegetative DSB repair nor definitively required for meiosis. What, then, is the functional significance of these structures?

Authors' response: Our data is consistent with an essential role for condensation in meiosis, but does not demonstrate that this is the case. Indeed, to show this, one would need a separation-of-function mutant that abolishes condensation alone. We do not have such a mutant, and finding one will be very difficult, if not impossible.

Revision Comment: This response is internally inconsistent with the earlier statement that “we do not claim that Mre11 condensation is essential in meiosis.”

While this reviewer appreciate the difficulty of generating a clean separation-of-function mutant, the absence of such evidence means that the functional relevance of Mre11 condensation remains entirely speculative. This returns directly to the original critique: what is the biological significance of Mre11 condensates?

A substantial portion of the manuscript is devoted to describing these condensates. Yet, without demonstrable functional consequences, these findings remain largely descriptive and fall short of the mechanistic or conceptual advance expected for a journal such as Nature Communications.

Authors' response: While we can't say why Mre11 condensation is dispensable in vegetative conditions, but (perhaps) important during meiosis, we speculate that the self-interactions provided by the Mre11 IDR facilitates its recruitment prior to DSB formation.

Revision Comment: The authors further speculate that Mre11 condensation may facilitate recruitment prior to DSB formation in meiosis, but this does not address the broader issue.

If condensate formation does not affect repair efficiency, survival, or measurable repair outcomes, then it is unclear why this phenomenon matters biologically. At present, the study does not convincingly explain why these condensates are more than an epiphenomenon of Mre11 self-association.

- This also raises additional concerns as to the importance of the Mre11 condensate concept in vegetative DSB repair. If it is not important, then what does it matter, or worse, who cares this condensate formation?

Authors' response: Our results imply that, whatever mechanism drives high local Mre11 accumulation following exogenous DNA damage in vegetative cells, it is not essential for DSB repair in that context. It is also possible that IDR-dependent focal accumulation might have functional consequences in DSB repair, but it simply did not manifest in our survival-based assay. Possibly, this accumulation could be a side effect of the self-assembly property of Mre11 that is important in other context, including meiosis. However, note that Mre11 also has established roles in other cellular contexts beyond exogenous DNA damage repair, including checkpoint activation, telomere maintenance, stabilizing stalled replication forks, etc. Mre11 condensation could potentially influence some of these pathways, though investigating this is beyond the scope of this study.

Revision Comment: This response again underscores the central problem: the manuscript does not establish a clear biological function for Mre11 condensates in any context tested. Invoking potential roles in other pathways without experimental support does not resolve this issue.

At present, the condensate concept remains insufficiently justified. Without functional evidence—either through clear phenotypic consequences, mechanistic linkage, or causal necessity—the biological significance of Mre11 condensation remains unclear.

Reviewer #4

(Remarks to the Author)

Dear Reviewers,

We thank the reviewers for their thoughtful and constructive comments on our manuscript entitled "Recruitment of Mre11 to recombination sites during meiosis". We have addressed all reviewers' comments in detail by performing additional experiments and provide a detailed point-by-point response below (reviewer comments are in black, responses are in blue).

Here is a summary of the major and minor experiments added to the manuscript:

Major experiments

1. To support our pulldown assays, we have performed yeast-2-hybrid experiments between Mre11-Mer2 and their respective mutants (Mer2-EQEK and Mre11-LLK residues). Our new data in Supplementary Fig. 6a confirms interaction between the wild-type proteins and show abolished Mre11-Mer2 interaction in the mutants.
2. To address concerns regarding the strength of interaction between Mre11 and Mer2, we performed biolayer interferometry experiments between Mer2 and the Mre11 C-terminal 49 residues, which confirmed this interaction is indeed weak (Supplementary Figure 6e). This supports the idea that this interaction is further stabilized through multivalency and SUMO-SIM interactions.
3. We further demonstrated by co-immunoprecipitation that Mre11 and Mer2 interact during meiosis and that Mer2-EQEK motif and Mre11 C-terminal helix are contribute to this interaction (Supplementary Figure 6j).
4. In response to the reviewers' concerns on the significance of SIM3 motif, we show by co-immunoprecipitation that SIM3 motif and C-terminal helix of Mre11 cooperate to promote interaction with Mer2 (Supplementary Figure 10f). This is consistent with the synergistic effect that we demonstrated previously by immunofluorescence, Southern blotting and spore viability analyses.

Minor experiments

1. We verified by immunofluorescence that the Mer2-EQEK mutant does not compromise Mer2 recruitment (Supplementary Fig. 6i).
2. We confirmed the Mre11-SIM3 and Mre11- Δ C15 mutations do not affect MMS-induced foci formation under vegetative conditions (Supplementary Fig. 10g).
3. We further showed that deletion of the Mre11 C-terminal helix abolishes Mer2-Mre11 co-localization *in vitro* (Supplementary Fig. 6d).

We thank the reviewers again for their valuable feedback on our manuscript.

Sincerely,

Corentin Claeys Bouuaert

REVIEWER COMMENTS

Reviewer #1 (Remarks to the Author):

This paper aimed to characterize the mechanism(s) by which the Mre11 complex (MRX in budding yeast) promotes meiotic DSB formation in budding yeast. It has long been known that Mre11 is required for meiotic DSB formation by Spo11, a function that does not involve Mre11 nuclease activity and that involves the last 49 aminoacids of Mre11.

Using a combination of in vitro biochemical and biophysical assays with recombinant proteins, AlphaFold modelling and yeast genetics, they bring novel insight into the functions of the Mre11 complex involved.

The main conclusions are that Mre11, as well as MRX, can form DNA-dependent condensates in vitro, like the authors previously showed for the other DSB proteins, Mer2 and Rec114-Mei4. Mer2 condensates promote the nucleation of Mre11 condensates. They better characterize the previously described interaction between Mre11 and Mer2, using AlphaFold and mutation of the predicted interacting residues. The interaction is important for meiotic DSB formation and spore viability, although it is the only function explaining Mre11 requirement for DSB formation. Finally, they uncover a novel SUMO-interaction motif in Mre11, which, in addition to the last alpha-helix of Mre11 is required for DSB formation. These results are complementary to recent findings that showed that Mre11 also directly interacts with Spo11, to promote DSB formation.

Altogether, although some results are still preliminary and the mechanism by which the MRX activates Spo11 is still not fully understood, this study brings novel important insight into the mechanisms by which the MRX promotes meiotic DSB formation, in particular through its interaction with Mer2. This study also reveals that meiotic DSB formation seems to be promoted, in vivo, by the formation of successive condensates, that may collaborate to concentrate Spo11 subunits and trigger its catalytic activity. Although the Mre11 complex is not required for meiotic breaks in mammals or plants, the general concept of condensates leading to meiotic DSB formation is of a broad interest.

Major comments:

- It would be important to examine Mer2 foci in vivo for the mer2-EQEK mutant, to measure if it is still forming normal foci. Especially important since its phenotype seems to be different from that of mre11 Δ C15.

Following the reviewer's suggestions, we analyzed the impact of the EQEK mutation on Mer2 foci formation by immunofluorescence and have observed no significant difference with the wild type (Supplementary Fig. 6i).

- In the same line, it would also be important to assess co-IP in meiotic cells between Mer2 and mre11 Δ C15 and between Mer2-EQEK and Mre11.

As suggested by the reviewer, we have performed co-IP experiments from meiotic extracts (4h) between Mre11 and the Mer2-EQEK mutant and between Mer2 and the Mre11- Δ C15 mutant (Supplementary Fig. 6j). We observed a strong reduction in interaction between Mre11 and Mer2-EQEK interaction and a moderate reduction between Mer2 and Mre11- Δ C15, in line with the stronger meiotic phenotype (DSBs and spore viability) of the Mer2-EQEK mutant compared to that of the Mre11- Δ C15 mutant.

To further verify the impact of the mutations on Mer2-Mre11 interaction, we also analyzed the Mer2-EQEK and Mer2-LLK mutants in yeast-2-hybrid experiments in vegetative cells, which confirmed that both mutations completely abolish the interaction between these proteins in this context (Supplementary Fig. 6a). This reinforces the notion that multiple mechanisms collaborate to stabilize the interaction between Mer2 and Mre11 in meiotic cells.

Finally, we demonstrate by co-IP that combining Mre11- Δ C15 with the Mre11-SIM3 mutation abolishes Mer2 interaction (Supplementary Fig. 10f), in line with a synergistic effect of these mutations on DSB formation.

- Figure S6A: the authors should repeat this experiment and add levels of a control protein, such as Pgc1, in order to say that mre11 Δ C15 levels were not changed. They do seem reduced here.

We repeated the experiment and included a Pgc1 control. The updated data is now shown as Supplementary Fig. 6f.

Other comments:

- In Figure S2A: why is the band corresponding to Xrs2 more intense than the two others? This would need some explanation.

The purified MRX complex contains a His-tag on Mre11 and a Flag-tag on Xrs2. The complex was purified by sequential affinity chromatography with Ni-NTA and anti-Flag resin, and partially dissociated during purification, yielding an excess of Xrs2. We have added this clarification in the figure legend.

- Figure S1C: the x axis seems to be wrong. It is rather time (minutes).

Fixed, thank you.

- Page 4 line 104: should be "Mre11 and MRX condensation...crowding agents (Figures 1E, 1F and S2B)"

Fixed, thank you.

- Figure 1B and 1D: it appears clearly that for the MRX, the total foci intensity is nearly zero after DNase treatment, whereas for Mre11, the total intensity is more or less preserved. So it does not seem that the remaining foci tend to accumulate more protein in the case of MRX. How to explain these results?

We thank the reviewer for raising this point and agree that the text did not accurately reflect that difference. We have now edited the text to more accurately describe the result.

The behaviour of Mre11 and MRX complexes in these assays should not be compared too closely because both the presence of Rad50 and Xrs2 and the different tags can alter the condensation dynamics. In this case, we are not sure what causes this difference. One possibility is that it is an indirect effect of different DNA-binding activities of the complexes that may cause different protective effects on DNaseI treatment. Other possibilities include that Mre11 alone has more hydrophobic surfaces exposed than the full complex, contributing to self-interactions. Given these uncertainties, we opted not to discuss these speculations and rather focus on the key results, which are that (1) condensation is DNA-dependent and (2) that the Mre11 C-terminus contributes to self-assembly.

- Figure 1J: focus number seems to be zero without treatment, because of the y axis scale. Can the authors give the actual numbers (in the text, or in the Figure legend)?

We have added the range of foci count to the Figure 1 legend.

- Page 5 line 124: "...important for meiotic DSB formation". It could be added that this region is also required for Mre11 binding to DSB hotspot (ref 40)

Added, thank you.

- Page 5 line 137 and Figure S3C, D and F: clarify what "Supershift" means ("complexes of reduced mobility"?)

We have included the following explanation in the legend of Supplementary Fig. 3. 'In panels C, D and F, a well-defined band is observed, suggestive of a stoichiometric complex ('Complex'). With some constructs, species of lower electrophoretic mobility are detected at high protein concentration, suggestive of higher-order oligomeric species ('supershift'). These oligomeric species depend on the Mre11 C-terminal IDR.'

- Page 5 line 151 "...dissolved MMS-induced foci" and Figures 3A and B: the authors could remind that this is after 4-5 minutes 1,6-hexanediol treatment (as indicated in the methods section)

Done.

- Page 6 line 175 and Figure 4E: the authors should show the data also for mer11 Δ C49

Added, thank you.

- Figure S4F: in the absence of quantification, this experiment should be remade since there is almost no Mre11-Myc signal in the 0h time-point.

We have repeated the experiment, and the updated data are now included in Figure S4F.

- Page 7 and Figure 4F: for the Δ IDR and the Δ C49 it seems difficult to assess the colocalization since Mre11 foci are almost absent.

We thank the reviewer for pointing this out. To improve clarity, we have added a new set of images in Supplementary Fig. 4h for Mre11-WT, Mre11- Δ IDR, and Mre11- Δ C49 in the presence of Mer2, with increased brightness of the eGFP channel for better visualization of Mre11. This confirms that the co-localization between Mer2 and Mre11 depends on the C-terminal 49 residues of Mre11.

- Page 6 line 208: "implicated in Mre11 binding": should refer to Figure S5B.

Added, thank you.

- Page 6 line 225: typo: should be "...Mer2 by approximately 45%"

Fixed, thank you.

- Page 9 line 271: "delayed and reduced DSB formation": from the two experiments shown in Figure 7B and Figure S10C, DSBs are delayed, but not reduced. This should be corrected.

Done.

- Figure S10C: the right title above the Southern blot should be mre11-SIM3 and not mre11- Δ SIM3

Fixed, thank you.

Reviewer #2 (Remarks to the Author):

This article builds on a previous study published in Nature in 2021, in which the authors reported that the yeast RMM protein complex assembles into nucleoprotein condensates along meiotic chromosomes, where it recruits Spo11 to stimulate DNA cleavage. In this new article, the authors characterize the role of the MRX complex in meiotic DSB formation. It was already known that the Mre11 C-terminus is required for meiotic DSB formation, and directly binds to several regions of Spo11 (as shown by pulldown and MST). Here, they show that in their in vitro conditions, Mre11 as well as the MRX complex are able to form condensates in the presence of DNA, and they identify that condensation relies on DNA binding but also on the presence of the Mre11 C-terminus that is not strongly involved in DNA binding but probably in uncharacterized protein-protein interactions. In cells, formation of meiotic foci of Mre11 depends on other DSB proteins. The authors having already shown that the RMM protein complex assembles into condensates, they tested if these condensates can recruit Mre11. They found that indeed, in cells with defective RMM condensation, the formation of Mre11 foci is strongly reduced. Moreover, they detect by yeast 2-hybrid a direct interaction between the RMM protein Mer2 and Mre11, and propose that this interaction is responsible for recruiting Mre11 in RMM condensates. Similarly, when expressing a Mre11 mutant deleted from all its C-terminal IDR region, the formation of Mre11 foci

is strongly reduced. Deletion of the Mre11 C-terminus known to favor MRX condensation only decreases the number of MRX foci but not their intensities in cells. It also decreases colocalization with Mer2. The authors propose that the direct interaction between Mer2 and Mre11 mediates the nucleation of MRX condensates, whereas the propensity of MRX to form condensates is responsible for the growth of the condensates. Finally, the authors try to understand how Mre11 binds to Mer2, based on AlphaFold calculations, pull-down assays and immunofluorescence experiments. They study how much sumoylation regulates this interaction, using NMR, ITC, and immunofluorescence experiments.

Figure 1 clearly shows that, in the tested conditions, Mre11 and MRX form condensates and that DNA favors nucleation of these condensates. In order to understand the molecular mechanisms of condensation, the authors tested the role of the different Mre11 regions in Figure 2. They first identified the different regions of Mre11. Therefore, in panel A, they present an AlphaFold Model of yeast Mre11. As the experimental 3D structure of this protein is available (PDB ref: 9BI4), it would be more reliable to present this structure. Moreover, in panel B, the authors present a prediction of the disorder as a function of the Mre11 sequence. Here again, from the analysis of the experimental structure, it is possible to identify the disordered regions of Mre11 bound to Rad50 and DNA: the cryo-EM structure of Mre11 is well-defined between residues 1 and 412 (phosphodiesterase and capping domains), and then between residues 442 and 522 (Rad50 binding domain). Such an analysis provides a precise definition (more reliable than a prediction) of the IDRs of yeast Mre11.

We thank the reviewer for the suggestion. We aligned the AlphaFold model of yeast Mre11 provided in Figure 2A to the cryo-EM structure of dsDNA bound Mre11-Rad50 complex and observed RMSD values of 0.99 Å and 1 Å for the two Mre11 molecules in the dimer, confirming strong agreement. Below we provide an aligned image of our model and the experimental structure to illustrate this.

(Left) Mre11 domains are colour-coded the same as in Figure 2A. The cryo-EM structure of Mre11 is in pale pink. The second Mre11, both Rad50 molecules, and the DNA are removed for clarity.

(Right) Cryo-EM structure of dsDNA bound Mre11-Rad50 complex¹. Both Mre11 molecules are in pale pink and magenta; Rad50 and DNA are in pale blue.

The conclusion is that all of the domains identified in Figure 2A are in agreement with the cryoEM structure. We chose to show the AlphaFold model instead, because it shows the C-terminal IDR and C15 alpha helix, which are the major focus of the manuscript. We have included a reference to the cryoEM structure to the paper though (line 112 in the main text and line 1006 in the figure legend of Fig. 2a).

In Figure 2 panels C, D and E, the authors show that deleting most of the phosphodiesterase domain (Del 10-270) or the capping domain, the linker between the capping and Rad50 binding domains, and part of the Rad50 binding domain (Del 290-472) decreases the number of foci, whereas deleting the C-ter IDR or the C-ter 49 last residues decreases the intensity of foci (but strongly increases their number) in vitro. So, the folded domains contribute to foci nucleation, whereas the C-ter IDR contributes to the growth of the condensates. When the same experiment is performed with the whole MRX complex, deletion of the C-ter 49 last residues decreases both the intensity and the number of foci, so that it seems that the C49 region is involved in both the nucleation and the growth of the condensates. Can the authors comment on that?

Indeed, the Mre11 C-terminus can be involved in both nucleation and growth.

First, as noted in response to reviewer 1 above, we think that the behaviour of Mre11 and MRX complexes should not be compared too closely. Overall, we find that deleting the Mre11 C-terminus reduces both foci number and intensity. Below a certain level of intensity, the foci count becomes more variable because it becomes more dependent on an arbitrary threshold. This is evident in the quantification of panel D, for instance where the error bars in foci numbers become bigger with the IDR and C49 truncations. Given this variability, we do not consider these numerical values to be very meaningful.

Second, foci number and intensity do not equate nucleation and growth, because both metrics can be impacted by phenomena that are only indirectly related to nucleation and growth per se. For instance, a mutation that reduces self-interaction may lead to higher foci number because the protein distributes more on DNA, but this doesn't mean that nucleation itself is enhanced.

Foci number is also a function of their rate of fusion but can be impacted by whether DNA sticks out of the condensates, which can act as a shield.

More generally, I wonder how it could be checked that the truncated variants of Mre11 are soluble, especially in the absence of Rad50. In Figure S3B, the authors show a convincing CBB picture of these different variants that have been successfully purified. However, truncating part of the folded domains, as in Del 10-270 and Del 290-472, creates exposed hydrophobic residues that might favor Mre11 aggregation, thus modifying the condensation mechanisms.

All the Mre11 truncations presented in this paper are purified under identical conditions and are analyzed by size-exclusion chromatography. None of the Mre11 truncations eluted in the void volume, indicating that they are not aggregated. Hence, there is no issue with solubility of the complexes per se. We agree that the truncations can impact condensation in various ways (perhaps through exposure of hydrophobic residues). Nevertheless, this does not affect the main conclusion of the analysis, which is that Mre11 condensation depends on its intrinsically disordered region.

Also, which interactions between Mre11 monomers could contribute to foci nucleation and growth? Does Mre11 dimerization contribute to the multivalent interactions involved in condensation ?

We thank the reviewer for raising this point. Indeed, we expect dimerization to contribute to Mre11 condensation.

Our data shows that the Mre11-IDR is necessary, but not sufficient for condensation (Figure S3E). Our interpretation is that the N-terminal domains contribute two key interactions: DNA binding (that provides anchor points) and dimerization (that contributes to multivalency). The IDR further contributes to DNA binding and self-interactions. In the absence of the Mre11 N-terminus, these are presumably too weak to support condensation. However, in the presence of Mer2 condensates, Mer2 can act as nucleation points, allowing Mre11-C49 peptides alone to accumulate through self-interactions (Figure 4G).

We have clarified these ideas in the discussion (Lines 317-326).

At this point, as the presence of DNA and the folded domains contribute to the nucleation of condensates, the authors characterize the DNA binding properties of their different Mre11 variants using EMSA. Here, as a non Mre11 expert, I don't understand how Mre11 binds to DNA. Indeed, in the cryoEM structure of Mre11/Rad50/dsDNA (PDB ref: 9BI4), Mre11 does not contact DNA. Moreover, yeast Mre11 is an acidic protein (IP around 5), only its C-terminal IDR is positively charged (the IP of its 49 Cter residues is particularly high: around 12). In the cryoEM structure of Mre11/Rad50/dsDNA, there are a few small positively charged patches at the surface of Mre11, however they are at the interface with Rad50. Was it previously reported that yeast Mre11 interacts with DNA? Does this happen in the activated state of Mre11? But then, is the interaction with Rad50 important for Mre11 to bind to DNA? Can the authors comment this point?

We thank the reviewer for this insightful point. Indeed, while it is known for several years that yeast Mre11 directly interacts with DNA *in vitro*² currently, no cryo-EM structure of yeast Mre11 or the MR sub-complex in the active state is available. However, a recent study³ solved the cryo-EM structure of the dsDNA-bound MR subcomplex in the catalytic state of Mre11 in *E. coli*. In this structure, the resting state shows Mre11's nuclease site blocked by ATP-Rad50, with flexible Rad50 coiled coils. Upon DNA binding, the coiled coils zip into a rod, forming a clamp around dsDNA, while Mre11 moves to the side, binds the DNA end, and assembles a nuclease channel. Based on this, Nicolas, *et al.*⁴ proposed an AlphaFold2 model of the DNA-bound catalytic state of yeast Mre11-Rad50, which resembles the *E. coli* structure, and provide evidence that this configuration depends on Sae2. While *in vitro* EMSA experiments show that Rad50 is not required for binding of Mre11 to DNA, Mre11-Rad50 oligomerization is required for proper DNA-binding behaviour and Mre11 endonuclease activity⁵.

In Figure 4, the authors observed the Mre11 foci in meiosis. They found that the 49 Cter residues are important for foci nucleation, which was not observed *in vitro*. Thus, additional cellular partners account for this nucleation effect. They propose that the RMM complex is able to trigger nucleation of Mre11 foci. To test this *in vitro*, they produced condensates of Mer2 and Mre11, and found, by mixing these condensates, that the two proteins co-localize. This colocalization is mildly favored in the presence of the folded domains, but strongly depends on the presence of the IDR region of

Mre11, and in particular its 49 Cter residues. How do the authors explain that the Mre11-C49 construct accumulates around Mer2 foci, but does not enter into these foci?

Our interpretation is that the full-length Mre11 protein enters Mer2 foci because it produces networks through protein-DNA interactions, Mre11 dimerization domain, plus Mre11-Mre11 interactions and Mre11-Mer2 interactions. The C49 residues lack a key DNA-binding and dimerization domain and seem to maintain only two properties: direct interaction with Mer2 and self-interaction between C49 peptides. Our understanding is that binding to Mer2 allows Mre11-C49 to locally reach a critical concentration that allows Mre11 accumulation through self-interaction. Presumably, Mre11-C49 does not enter Mer2 foci because their homotypic interactions are higher than heterotypic interactions with Mer2.

The above explanation is added to lines 317-326 of Discussion.

In Figure 5, the authors attempt to obtain a structural description of the interface between Mre11 and Mer2. They provide an AlphaFold2 model of the complex between 4 Mer2 molecules and 4 Mre11 C49 peptides. Indeed, they showed in a previous paper that Mer2 is a coiled-coil protein forming a tetramer (based on SEC-MALS and SAXS data). I found no Material and Methods section related to the AlphaFold2 calculations, and no scoring of the models (pLDDT plots; TM and ipTM scores). In Figure S5A, the heat maps suggest that the predictions are unreliable. Moreover, AlphaFold2 is known to poorly predict coiled-coil containing structures. From my point of view, if the AlphaFold2 models are poorly scored, they should not be shown. The authors can make hypotheses on the interfaces independently of the AlphaFold2 results. But showing / discussing AlphaFold2 models that are provided with low confidence scores is misleading: the reader is tempted to believe in these models, forgetting that they are based on no data. I understand that the authors found, by analyzing their models, that the motif SSM1 of Mre2 is involved in Mre11 binding, and this was previously shown elsewhere. But models obtained with poor scores do not reinforce the experimental data. As the authors can purify Mer2 and buy a synthetic peptide of Mre11 corresponding to C49, why didn't they use SEC-MALS and SAXS to characterize the Mer2-Mre11 complex, as previously reported for Mer2 alone? Is it because "the interaction is relatively weak" ? What do "relatively weak" mean ? Could the authors use ITC or BLI to measure the affinity between Mer2 and Mre11 ?

We thank the reviewer for raising the concern regarding the presented AlphaFold models. We have included ipTM scores for the models in Table S6. While we agree with the reviewer that the AlphaFold models have low confidence scores, our intention behind adding the models was to show a hypothesis-driving illustration to highlight the potential spatial arrangement of Mre11-C49 in presence of tetrameric Mer2. Indeed, the AlphaFold models alone do not form the basis of a mechanistic conclusion.

Following the reviewer's suggestions, we performed BLI experiments of Mre11-C49 and Mer2 coiled-coil and found a K_D value of 1.16 mM, indicative of a weak interaction between both the proteins. This data is added to Supplementary Figure 6e.

Note that this weak interaction does not undermine the biological importance of this interaction. First, in the context of the full-length protein, the interaction will be reinforced by the dimeric stoichiometry of Mre11. Second, we show that during

meiosis, this interaction works together with a SUMO-SIM interaction to allow Mre11 recruitment.

We also attempted ITC experiments between Mre11-C49 with Mer2 coiled-coil. Titration of Mre11-C49 into Mer2 produced endothermic peaks that also appeared when the peptide was titrated into buffer, rendering the results inconclusive (see below).

These data indicate that the interaction is indeed weak, as we had deduced from the pulldown analyses. Biophysical characterizations via SEC-MALS and SAXS are therefore not feasible.

In order to identify the residues at the interface between Mer2 and Mre11, the authors present pulldown assays revealed by WB. Could it be added to the legend of Figure 5B that the input and pulldown are revealed by WB using Ab against MBP? It is in the Material & Methods section, but it is essential for understanding the figure panel.

Added to the legend, thank you.

There are a lot of not reproducible interaction data obtained by pulldown revealed by WB in the literature. To keep this data as a main figure panel, I would recommend to confirm this experiment using another experimental method. As for ex by BLI, coating a synthetic biotinylated Mre11-C49 peptide on streptavidin beads and adding increasing concentrations of HisSUMO-Mer2. Or by MST using a fluorescent Mre11-C49 peptide.

We thank the reviewer for the suggestion. To complement our pulldown experiments, we performed the following additional experiments:

- BLI: Mre11-C49 and Mer2 coiled-coil interaction show a K_D of 1.16 μM, indicative of a weak interaction (Supplementary Fig. 6e).
- Yeast-2-hybrid: Interaction between Mer2 and Mre11-LLK and between Mre11 and Mer2-EQEK was abolished (Supplementary Fig. 6a).
- Co-immunoprecipitation in meiotic cells: We observed interaction between Mre11 and Mer2, while Mre11-ΔC15 and Mer2 and Mer2-EQEK and Mre11 showed reduced interaction (Supplementary Figs. 6j, 10f).

As sumoylation sites were mapped to the SSM1 region of Mer2, potentially involved in Mre11 binding, the authors searched for the impact of sumoylation on Mer2-Mre11 binding in cells. They detected 3 SUMO-interacting motifs in Mre11, two in the phosphodiesterase domain and one in the Cter IDR, just before the last 49 residues

(633-637). They used AlphaFold2 to predict that yeast Smt3 binds to the third SIM (SIM3). Here again, no scoring is provided to judge about the robustness of the prediction. However, a careful NMR analysis (though at low salt concentration) demonstrated that the SIM3 peptide specifically binds to the SIM-binding groove of Smt3. The ITC analyses provided the K_d of the interaction. In Figure S9, the information on concentration is misleading: only one concentration is written so that it is unclear if this is the concentration of the Smt3 protein or the peptide. Could the authors add the control obtained by injecting the WT peptide into the buffer?

Thanks for these suggestions. iPTM and pTM scores for AlphaFold3 predictions are now included in the legend of Figure S7D. Additionally, integrated heat peaks for the injection of WT SIM peptide and mutant SIM peptide into buffer are now added as panels (D) and (E) respectively of Figure S9. We have clarified the concentrations on the figure and in the legend.

In general, as there is no proof that in cells SIM3 SUMOylation is important for meiosis, and as the impact of sumoylation of SIM3 on the interaction between Mer2 and Mre11 is still unclear, the analysis of the impact of sumoylation does not strengthen this study and could be moved to the supplementary figures.

We respectfully disagree with this comment. The initial version of our manuscript already established that the SIM3 motif of Mre11 works together with the C-terminal helix to promote Mre11 recruitment to chromatin, and that both are important for DSB formation.

Nevertheless, to reinforce our claim that SIM3 is relevant for Mre11 recruitment during meiosis, we performed co-immunoprecipitation experiments between Mre2 and Mre11 in meiotic yeast extracts. These show that mutating the Mre11 SIM3 motif and deleting the C-terminal alpha helix reduce the interaction with Mer2. Importantly, combining the mutations abolish the interaction entirely (new data added to Supplementary Figure 10f). These results are in agreement with our IF, Southern blot, and spore viability data that showed a synergistic effect of these mutations.

Discussion:

In general, the discussion is based on experimental data that are still to be strengthened. When discussing about "the Mer2 alleles previously reported to reduce Mre11 interaction that involve amino acids predicted to point inside the Mer2 tetrameric coiled coil", why should Mer2 stay in its tetrameric state upon binding to Mre11 ?

The point we were making is that, based on the AlphaFold model, the C-terminal helix of Mre11 engages simultaneously to two adjacent Mer2 helices. Hence, mutating residues inside of the hydrophobic bundle could prevent Mre11 binding by destabilizing the coiled coil. Hence, we think the mutations we identified more directly impact the interaction than the previously-reported alleles.

Minor points:

1. Condensation: in the Mat & Met, the conditions used to study condensation include PEG (these conditions are precisely: 25 mM HEPES-HCl (pH 7.5), 2 mM DTT, 1 mg/ml

BSA, 5% glycerol, 5 mM MgCl₂, 5% PEG-3350, and 120 mM NaCl), whereas in the legend of Figure 1E,F, the impact of adding PEG is shown. Is there PEG in the dilution buffer only for the experiments displayed in Figure 1E,F ?

All condensation assays included 5% PEG in the reaction buffer, unless mentioned in the legends. Experiments shown in figure 1E, F contained 5% PEG and the control reactions were performed without PEG. We have clarified it in the figure legends, thank you.

2. NMR: in the last sentence of the Mat & Met sections, there is an n value that is cited but not present in the equation used for the CSP analysis.

Thank you, the mistake has been corrected.

3. NMR and ITC: why is the salt concentration so different between the NMR and ITC conditions ? The ITC conditions are closer to physiological conditions than the NMR conditions. Is the salt concentration influencing the interaction?

Thank you for pointing that out. Indeed, a lower salt concentration was used for NMR to improve signal quality. Higher salt was used for ITC to determine K_d values closer to physiological conditions.

Reviewer #3 (Remarks to the Author):

NCOMMS-25-61398-T

Recruitment of Mre11 to recombination sites during meiosis
In this manuscript, Priyadarshini et al. reported that a widely studied DSB end processing factor, Mre11 of the MRX complex, forms condensate at DSB foci in both vegetative cells and during meiosis of *S cerevisiae*, and this involves a C-terminal IDR and the last C-terminal 15 amino acids that bind Mer2 and SUMO modification. Interestingly, while the condensate is dispensable for vegetative DSB repair, it is essential for meiosis. While some results are interesting and may offer new insights into the role of Mre11 in *cerevisiae* meiosis, the main conclusion lacks strong support and significant issues remain unanswered.

Major issues:

1. The biggest issue is the unclearness of the true biological significance of the Mre11 condensate in DSB repair.

- Why the condensate demonstrated different essentialities to vegetative DSB repair and meiosis? What is unclear is that if the condensates are part of the DSB foci and its formation involved the same protein-protein interaction principles (or same proteins and domains are involved), why are they essential for meiosis but not for DSB repair? What determines the different essentialities here?

Although the domains required for Mre11 foci formation in meiotic and vegetative conditions are the same, the assembly of these foci and their composition are fundamentally different. Vegetative Mre11 foci depend on DSBs, meiotic ones do not. Meiotic Mre11 foci depend on Mer2 and involve a host of other proteins that are absent from vegetative foci.

Note that we do not claim that Mre11 condensation is essential in meiosis. We claim that it cannot be essential in vegetative DSB repair (because deleting the IDR compromise condensation *in vitro* and foci formation *in vivo* with no detectable effect on MMS sensitivity).

Our data is consistent with an essential role for condensation in meiosis, but does not demonstrate that this is the case. Indeed, to show this, one would need a separation-of-function mutant that abolishes condensation alone. We do not have such a mutant, and finding one will be very difficult, if not impossible.

While we can't say why Mre11 condensation is dispensable in vegetative conditions, but (perhaps) important during meiosis, we speculate that the self-interactions provided by the Mre11 IDR facilitates its recruitment prior to DSB formation.

- This also raises additional concerns as to the importance of the Mre11 condensate concept in vegetative DSB repair. If it is not important, then what does it matter, or worse, who cares this condensate formation?

Our results imply that, whatever mechanism drives high local Mre11 accumulation following exogenous DNA damage in vegetative cells, it is not essential for DSB repair in that context. It is also possible that IDR-dependent focal accumulation might have functional consequences in DSB repair, but it simply did not manifest in our survival-based assay. Possibly, this accumulation could be a side effect of the self-assembly property of Mre11 that is important in other context, including meiosis. However, note that Mre11 also has established roles in other cellular contexts beyond exogenous DNA damage repair, including checkpoint activation, telomere maintenance, stabilizing stalled replication forks, etc. Mre11 condensation could potentially influence some of these pathways, though investigating this is beyond the scope of this study.

We have added the above explanation to lines 339-348 of Discussion.

- Then how to separate this condensate formation from the normal Mre11 foci at DSB repair site in vegetate cells?

We are not sure to really understand what the reviewer means here. What are the normal Mre11 foci the reviewer refers to here? The MMS-dependent foci, not MMS-induced? This is unclear.

Perhaps the reviewer is trying to ask what is the relationship between *in vitro* condensates and *in vivo* foci? As in, do the same biochemical principles that allow *in vitro* condensation applies *in vivo*? Of course, we cannot answer that question. We observe correlations (e.g. dependence on the IDR) but cannot be sure of the extent to which these assemblies are related.

- While a DSB repair assay was performed for meiotic cells, it was absent for vegetate cells. Only a survival result was shown for vegetate cells expressing Mre11 mutants. Should also examine if these condensate deficient Mre11 mutants impact DSB repair in vegetate cells.

We agree that it would be interesting to analyze the impact of these mutants in vegetative DSB repair in more detail. It is possible that these would uncover more subtle roles of the Mre11 IDR. However, this study focuses primarily on the meiotic function of Mre11, and such physical analysis of DSB repair are beyond the scope of this study.

- If condensate is essential for meiosis, can you rescue the lethality issue of the strains with Mre11-IDR or C49 deleted? Are there any analyses to see if the Mre11-IDR or the C49 residues alone can form foci in meiotic cells?

It is not clear what the reviewer is suggesting. Rescuing the lethality of which strains with Mre11-IDR or C49?

In theory, we could have tested whether Mre11-IDR or C49 can form foci in meiotic cells. We did not perform this experiment because we do not consider this a very likely scenario, because neither IDR alone nor C49 alone form condensates in vitro, even at relatively high concentration (i.e. they are necessary but not sufficient for condensation). In addition, although full-length Mre11 binds Mer2 efficiently in a yeast-two-hybrid assay, the C49 residues alone do not (Supplementary Fig. 6a). This is because individual C49 peptides have low affinity for Mer2 (Supplementary Fig. 6e). Presumably, the affinity of full-length Mre11 is higher because it forms a stable dimer that enhances the interaction.

Without clear answers to these questions, the true biological significance of the Mre11 condensates at DSB sites remain uncertain.

2. How exactly Mre11 condensates are formed, regulated, and contributed to DSB foci in vegetate and meiotic cells is unclear.

- First, based on data presented, there are five regions/motifs of Mre11 shown to be involved: the IDR (~520-677), the C49 (residues 643-692), the C15 (residues 677-692), and the SUMO binding region (residues 633-637). The C15 binds to Mer2 and deletion of which caused similar level of foci reduction as the C49 deletion; yet, C15 is not essential for meiosis essentiality, whereas the C49 does. These results suggest the last 34 residues (643-677) of the IDR are important for meiosis. However, a comprehensive comparison of these above five fragments on in vitro condensate assembly with or without Mer2 is lacking. Also, how these fragments contribute to Mre11 foci formation in vegetate vs meiotic cells are unclear.

Indeed, recently work (Aithal, 2024) shows that the last 34 residues of Mre11 are required for interaction with Spo11, as mentioned in the discussion.

We agree with the reviewer that each of the mutants and truncations studied in the manuscript need to be systematically characterized. In an effort to complete the analyses of the mutants, we purified eGFP-Mre11- Δ C15 and performed condensation assays with and without Mer2. New data is added to Supplementary Fig. 6b-d. Similar to eGFP-Mre11- Δ C49, we observed reduced foci intensity and colocalization with Mer2 foci, indicating that these residues contribute to condensate formation and interaction with Mer2 within condensates.

- Second, there is the lack of DSB repair in vegetative vs meiotic cells expressing Mre11 mutants with deletion of just the above five fragments. Similarly, meiotic cell viability of these mutants is important to show.

To further complete the characterization of each of our mutants, we performed immunofluorescence and MMS sensitivity analyses of Mre11- Δ C15, Mre11-SIM, and Mre11- Δ C15+SIM strains, which revealed no detectable effect in vegetative conditions, as expected. These data are presented in Supplementary Fig. 10g, h.

Yeast spore viability data for each of the mutants are provided in Figure 4C (Mre11-WT, Mre11- Δ C49, Mre11- Δ IDR) and Figure 7C (Mre11-WT, Mre11- Δ C15, Mre11-SIM, Mre11- Δ C15+SIM), confirming the impact of these fragments in meiosis.

3. While the manuscript presented some interesting findings, often there was no discussion regarding the biological/pathological implications/significance of the findings.

- Fig 3B has no discussion regarding the increase in the percentage of Mre11 aggregates by 1,6-HD treatment and the relationship of this increase to the reduction in Mre11 foci.

Lines 327-338 are added to the discussion.

- Fig 4A showed no Mre11 aggregates by 1,6-HD, which is different from vegetative cells and somehow may indicate a difference between these two states. But then, it did so when expressed in meiotic mer2-KRRR cells in Fig 4D. So, going back to the point above, what are exactly the difference between these foci and the large-in-size aggregates of Mre11? Are these reflecting real biological/pathological changes or just expression-induced nuance of Mre11 under different background? This also ties to the major issue #1.

In vegetative conditions, hexanediol treatment led to hyper-accumulation of Mre11 into one or a few large aggregates which also mirrored the aggregates observed in a sub-population of cells harboring mre11- Δ IDR or mer2-KRRR mutations. The reason for this hyperaccumulation remains unclear. It could be a consequence of the self-assembly property of Mre11 or the MRX complex, akin to synaptonemal complex proteins that form polycomplexes under pathological conditions or may reflect a binding preference of Mre11/MRX to an unknown partner. One way to understand this is that, under normal (vegetative or meiotic) conditions, the affinity of Mre11 to its substrate and cellular partners allow its localization throughout the nucleus. In some pathological conditions (e.g. upon hexanediol treatment in vegetative cells, or in a mer2-KRRR background in meiotic cells), the network of interactions that fine-tunes

Mre11 localization is compromised, leading to its local hyperaccumulation (by self-assembly or by tethering to unknown cellular factors). Nevertheless, they are not due to expression-induced artifacts, as TCA extractions show no obvious changes in Mre11 levels in the mutants strains or upon hexanediol treatment (Figure S4). We have broadly discussed this in lines 327-338 of Discussion.

Minor issues:

- Fig 1B & 1D: it stated that “While nuclease treatment strongly reduced the numbers of Mre11 and MRX foci, the minority of Mre11 foci that resisted nuclease treatment tended to accumulate more protein”. Where is the evidence that the residual condensate accumulated more protein? The selected representative images clearly did not support that. The quantitative intensity data in Fig 1B and 1D showed much reduced fluorescence intensities, which reflect the abundance of proteins in any condensate, for all foci (including the nuclease resisted ones). None of the quantitated foci after nuclease treatment retained an intensity level even close to the lowest data point prior to nuclease treatment in these two panels. This statement clearly does not hold.

We thank the reviewer for pointing this out. To clarify, we quantified intensity per focus in both untreated and DNaseI-treated Mre11 condensates, and the data are now shown in Supplementary Fig. 1f. The representative image has been updated to better reflect these quantifications. We also clarified in the main text that this accumulation is observed for Mre11-treated samples, but not MRX.

- Fig 4F was assembled with equal molar ratio of Mre11 and Mer2, then why a 4-fold ratio was used for Fig 4G (Mre11-C49:Mer2=4:1)? Will Mre11-C49 still coat Mer2 condensates if mixed at 1:1 ratio? In fact, if Mer2 works as a tetramer to recruit the MRX complex, it would be Mre11-C49:Mer2=1:4 in these in vitro assembly reactions.

This is not correct, since each Mer2 tetramer has four potential Mre11 binding sites (Figure S5A). Either way, we are not comparing the results of Figure 4F with Figure 4G, and a stoichiometric assembly is not essential when studying colocalization between Mre11 and Mer2 condensates because condensates are formed by multivalent, collective interactions that are not dictated by rigid stoichiometry.

- Fig 5A only modeled the C49 of Mre11. What will happen if a larger domain of Mre11 (like 620-692 used in Fig S7A) is used? This is important as the authors proposed multiple motifs of Mre11 binding to Mer2. There are a lot of back-and-forth information regarding the domains that are involved in Mer2 binding and condensate formation of Mre11, which were not clearly and systematically evaluated.

To clarify, we do not report multiple interaction domains between Mer2 and Mre11. We report (i) a direct protein-protein interaction between the Mre11-C15 helix (involving LLK residues) that binds to Mer2-SSM1 (involving EQEK residues), and (ii) a SUMO-SIM interaction between Mre11-SIM3, and an unidentified SUMOylated target (that may be Mer2). In addition, we also propose that (iii) Mre11 self-interaction (through C-term IDR and C49) may participate in Mre11 recruitment and accumulation during meiosis.

Modelling the Mer2-Mre11 and SUMO-SIM interaction simultaneously is not feasible because AlphaFold3 doesn't allow incorporation of SUMO modifications.

- Fig 7: The importance of the SIM3 motif in Mer2 binding in vivo is lacking. Why not performing similar pulldown assay as in Fig 5B for Mre11-SIM3 mutant and Mer2? Without this information, the model in Fig 7D and the conclusion in the text are premature.

We disagree with the reviewer. The initial submission presented detailed characterization of the SIM3 motif in vitro (NMR and ITC, Figure 6) and in vivo (IF, Southern blotting and spore viability, Figure 7). These data indicate that the SIM3 has a subtle role in meiosis, but collaborates with the C-terminal helix in recruiting Mre11. In the revised manuscript, we further reinforced this conclusion by including co-IP data (Supplementary Fig. 10f) that also confirms the synergistic effect of the SIM3 and C15 deletion on Mer2 interaction. Therefore, we are confident about our conclusions and the model presented in Figure 7D.

- Some of the selected representative images are not impressive to support the conclusions. For instance, Fig 1A and S1C hardly tell any foci intensity increases. Fig 2C did not show any reduction in foci number for Mre11 Δ 290-472. The IDR deletion Mre11 mutant had much weaker foci than the C49 deletion in Fig 4B; yet, the quantitation showed the opposite.

We understand that this is somewhat confusing, but please note that the quantification in Figure 4B shows foci number, not intensity. The images shown are indeed representative of the results, because the foci observed in Mre11- Δ IDR mutants are much less intense as those observed for Mre11-C49, as described in the text.

- Figure legends are generally too simplified with much information missing. For instance, what was the conc of the eGFP-Mre11 protein in Fig S1B? What are conc of DNA in Fig 1A and 1C? What time point was the image taken for Fig 1A? Do Fig 2C-E contain a crowding agent? It would help the readers if all figure legends provided detailed information.

The experimental conditions of the condensation assays are detailed in the M&M section. All reactions are performed in identical standard conditions, unless specified. We have verified all the legends and made sure that the relevant information is provided when the experimental condition differed from the standard condition.

- "...focus formation of Mer2-EQEK was diminished compared to wild-type Mre11 by approximately 45% (Figure 5D)": Fig 5 D has nothing to do with Mer2 foci, but instead, it showed Mre11 foci under different backgrounds including in the Mer2-EQEK strain.

We thank the reviewer for pointing that out. This mistake has been corrected.

Reviewer #4 (Remarks to the Author):

References in support of the point-by-point response:

1. Hohl, M., Yu, Y., Kuryavyi, V., Patel, D. & Petrini, J. Structure guided functional analysis of the *S. cerevisiae* Mre11 complex. *Nature Communications* **16**, 7469 (2025).
2. Furuse, M. *et al.* Distinct roles of two separable *in vitro* activities of yeast Mre11 in mitotic and meiotic recombination. *The EMBO Journal* **17**, 6412–6425 (1998).
3. Käshammer, L. *et al.* Mechanism of DNA End Sensing and Processing by the Mre11-Rad50 Complex. *Molecular Cell* **76**, 382-394.e6 (2019).
4. Nicolas, Y. *et al.* Molecular insights into the activation of Mre11-Rad50 endonuclease activity by Sae2/CtIP. *Molecular Cell* **84**, 2223-2237.e4 (2024).
5. Kissling, V. M. *et al.* Mre11-Rad50 oligomerization promotes DNA double-strand break repair. *Nat Commun* **13**, 2374 (2022).

Dear Reviewers,

We thank the reviewers for their constructive comments on our manuscript entitled "Recruitment of Mre11 to recombination sites during meiosis". Point-by-point responses are below (reviewer comments are in black, responses are in blue).

Sincerely,

Corentin Claeys Bouuaert and Priyanka Priyadarshini

REVIEWERS' COMMENTS

Reviewer #1 (Remarks to the Author):

In this revision, the authors have satisfactorily answered all my major and minor points.

Thank you.

Reviewer #2 (Remarks to the Author):

The authors answered to all my questions, I am fine with their answers, thanks. Except for the addition of the BLI experiments (Suppl. Fig. 6e). For me, it is mandatory to delete this panel. First, I don't see how you can measure an affinity of 1 nM by titrating a ligand between 12.5 and 25 μ M. Second, several essential controls are lacking, as titrating the analyte onto empty biosensors (Nickel-biosensors interact non-specifically with many ligands) and onto His-SUMO coated biosensors. Third, the legend is unclear, but, if, as I understood, the only displayed control experiments gave the less bright curves (titration onto the His-SUMO-Mer2 coated chip of MBP alone), then there is a huge non specific signal that should be subtracted from the main BLI curve.

For all these reasons, I think that this panel has to be removed.

We agree with the suggestion of Reviewer 2 and have removed the BLI data in Supplementary Fig. 6e.

Reviewer #3 (Remarks to the Author):

While the authors have addressed many technical points, my central concern remains largely unaddressed: the manuscript still fails to establish the biological significance of Mre11 condensates. Below, I focus specifically on this unresolved issue and the authors' responses to it.

Original comment: The biggest issue is the unclearness of the true biological

significance of the Mre11 condensate in DSB repair.

- Why the condensate demonstrated different essentialities to vegetative DSB repair and meiosis? What is unclear is that if the condensates are part of the DSB foci and its formation involved the same protein-protein interaction principles (or same proteins and domains are involved), why are they essential for meiosis but not for DSB repair? What determines the different essentialities here?

Authors' response: Although the domains required for Mre11 foci formation in meiotic and vegetative conditions are the same, the assembly of these foci and their composition are fundamentally different. Vegetative Mre11 foci depend on DSBs, meiotic ones do not. Meiotic Mre11 foci depend on Mer2 and involve a host of other proteins that are absent from vegetative foci.

Note that we do not claim that Mre11 condensation is essential in meiosis. We claim that it cannot be essential in vegetative DSB repair (because deleting the IDR compromise condensation in vitro and foci formation in vivo with no detectable effect on MMS sensitivity).

Revision Comment: If the authors do not claim that Mre11 condensation is essential in meiosis, then what is the intended biological conclusion regarding these condensates? Under the authors' own framing, Mre11 condensation is neither essential for vegetative DSB repair nor definitively required for meiosis. What, then, is the functional significance of these structures?

Authors' response: Our data is consistent with an essential role for condensation in meiosis, but does not demonstrate that this is the case. Indeed, to show this, one would need a separation-of-function mutant that abolishes condensation alone. We do not have such a mutant, and finding one will be very difficult, if not impossible.

Revision Comment: This response is internally inconsistent with the earlier statement that "we do not claim that Mre11 condensation is essential in meiosis."

While this reviewer appreciate the difficulty of generating a clean separation-of-function mutant, the absence of such evidence means that the functional relevance of Mre11 condensation remains entirely speculative. This returns directly to the original critique: what is the biological significance of Mre11 condensates?

A substantial portion of the manuscript is devoted to describing these condensates. Yet, without demonstrable functional consequences, these findings remain largely descriptive and fall short of the mechanistic or conceptual advance expected for a journal such as Nature Communications.

Authors' response: While we can't say why Mre11 condensation is dispensable in vegetative conditions, but (perhaps) important during meiosis, we speculate that the self-interactions provided by the Mre11 IDR facilitates its recruitment prior to DSB formation.

Revision Comment: The authors further speculate that Mre11 condensation may facilitate recruitment prior to DSB formation in meiosis, but this does not address the broader issue.

If condensate formation does not affect repair efficiency, survival, or measurable repair outcomes, then it is unclear why this phenomenon matters biologically. At present, the study does not convincingly explain why these condensates are more than an epiphenomenon of Mre11 self-association.

- This also raises additional concerns as to the importance of the Mre11 condensate concept in vegetative DSB repair. If it is not important, then what does it matter, or worse, who cares this condensate formation?

Authors' response: Our results imply that, whatever mechanism drives high local Mre11 accumulation following exogenous DNA damage in vegetative cells, it is not essential for DSB repair in that context. It is also possible that IDR-dependent focal accumulation might have functional consequences in DSB repair, but it simply did not manifest in our survival-based assay. Possibly, this accumulation could be a side effect of the self-assembly property of Mre11 that is important in other context, including meiosis. However, note that Mre11 also has established roles in other cellular contexts beyond exogenous DNA damage repair, including checkpoint activation, telomere maintenance, stabilizing stalled replication forks, etc. Mre11 condensation could potentially influence some of these pathways, though investigating this is beyond the scope of this study.

Revision Comment: This response again underscores the central problem: the manuscript does not establish a clear biological function for Mre11 condensates in any context tested. Invoking potential roles in other pathways without experimental support does not resolve this issue.

At present, the condensate concept remains insufficiently justified. Without functional evidence—either through clear phenotypic consequences, mechanistic linkage, or causal necessity—the biological significance of Mre11 condensation remains unclear.

Following the comments of Reviewer 3 on the biological relevance of condensation, we have toned down the Discussion on the same.

Reviewer #4 (Remarks to the Author):
